# Gastrin-releasing peptide signaling in the nucleus accumbens medial shell regulates neuronal excitability and motivation

Erin E. Aisenberg [1,2], Thomas L. Li[2,3], Hongli Wang[2,3], Atehsa A. Sahagun [1,2], Emilie M. Tu [3] & Helen S. Bateup [1,2,3] ✉

Neuropeptides are the largest class of neuromodulators. It has been shown that subpopulations of ventral tegmental area (VTA) dopamine neurons express mRNA for the neuropeptide Gastrin-releasing peptide (GRP); however, its functional relevance in mesolimbic circuits is unknown. Here we report that the GRP receptor (GRPR) is expressed in nucleus accumbens medial shell (NAc MSh) neurons, which are targeted by GRP-expressing inputs from the VTA, hippocampus, and amygdala. We show that NAc MSh GRPR-positive cells represent subpopulations of D2 receptor-expressing neurons, comprising both classical striatal projection neurons (SPNs) and eccentric SPNs. GRPR-expressing neurons have high intrinsic excitability and can be activated by GRP in vivo. NAc-specific deletion of *Grpr* in mice increases motivation in a progressive ratio test, demonstrating that GRPR regulates motivated behaviors. These experiments establish GRP/GRPR signaling as a potent modulator of mesolimbic circuits and advance our understanding of the diversity of cell types present in the NAc.

Neurons have traditionally been defined by their expression of classical neurotransmitters, such as glutamate and GABA. However, it has become appreciated that certain populations of neurons can co-release more than one type of neurotransmitter[1], suggesting increased signaling complexity. In addition to classical neurotransmitters, many neurons also express genes that encode for neuropeptides—small short-chain polypeptides that act via metabotropic G-protein-coupled receptors (GPCRs)[2]. Neuropeptides exert potent influence over neuronal excitability, synaptic transmission, plasticity, and gene expression programs[3]. Single-cell RNA sequencing (scRNA-seq) analyses have revealed that many neurons have the genetic machinery to synthesize and respond to neuropeptides[4], but their actions have been relatively understudied in the brain.

A previous scRNA-seq study of mouse midbrain dopamine (DA) neurons found that ~30% of DA neurons in the ventral tegmental area (VTA) and the ventral substantia nigra pars compacta (vSNc) express *Grp* mRNA[5]. *Grp* encodes Gastrin-releasing peptide (GRP), a bombesin-like neuropeptide that signals through a $G_q$-coupled GPCR (gastrin-releasing peptide receptor, GRPR, also called BB2)[6,7]. GRP was originally named for its role in stimulating gastrin release from G cells in the stomach[8]. In the central nervous system, GRP has been studied in the spinal cord in the context of itch and pain processing[9–11]. GRP and GRPR are expressed relatively sparsely in the brain[12]. However, studies have demonstrated a role for GRP/GRPR signaling in emotionally-motivated and fear memories as well as responses to stress[13,14]. In the midbrain, prior studies demonstrated GRP immunoreactivity in the SNc[15]. Additionally, *Grp* was identified as a differentially expressed gene between VTA and SNc DA neurons[16,17]. *GRP*-expressing DA neurons have also been observed in post-mortem human brains and are suggested to be relatively resistant to degeneration in Parkinson's disease[16]. Given the presence of GRP in behaviorally- and disease-relevant nigrostriatal and mesolimbic projections, we sought to define the functional relevance of GRP-GRPR signaling within these circuits.

[1]Helen Wills Neuroscience Institute, University of California Berkeley, Berkeley, CA, USA. [2]Department of Neuroscience, University of California Berkeley, Berkeley, CA, USA. [3]Department of Molecular and Cell Biology, University of California Berkeley, Berkeley, CA, USA. ✉e-mail: bateup@berkeley.edu

Circuit mapping experiments indicated that *Grp*-expressing DA neurons project to regions within the dorsal and ventral striatum, specifically the nucleus accumbens medial shell (NAc MSh)[5], where GRP may influence the activity of downstream neurons. The striatum is the main input structure of the basal ganglia, which integrates glutamatergic inputs from the cortex and thalamus, DA from the midbrain, GABAergic inhibition from multiple intra-striatal sources, and acetylcholine from cholinergic interneurons[18]. GABAergic striatal projection neurons (SPNs) are the major output neurons of the striatum, comprising 90-95% of the neuronal population[19]. SPNs are canonically divided into direct-pathway and indirect-pathway neurons based on their primary projections to either the substantia nigra pars reticulata and globus pallidus internal segment or globus pallidus external segment, respectively[20]. Direct pathway SPNs (dSPNs) typically express type 1 DA receptors (D1) while indirect pathway SPNs (iSPNs) generally express type 2 DA receptors (D2) and adenosine 2 A receptors (A2A)[20]. A third type of SPN, eccentric SPNs (eSPNs) was recently proposed[21]. eSPNs are defined by their expression of SPN markers, including *Ppp1r1b*, *Adora2a*, and *Drd1,* as well as a unique gene module consisting of ~100 genes[21]. Unlike dSPNs and iSPNs, nothing is known about the functional properties of eSPNs and there are currently no mouse lines that selectively label them.

The dorsal striatum (DS) is important for motor functions, including action selection and motor learning, while the ventral striatum (VS) controls motivated behaviors and reward learning[22]. In the DS, activation of dSPNs with optogenetics promotes locomotor activity while bulk activation of iSPNs inhibits movement[23,24]. During normal movement, however, both pathways are active and work in a coordinated manner[25,26]. The VS also contains dSPNs and iSPNs. In the NAc, the dSPNs have canonically been thought to directly project to and inhibit the VTA, promoting motivated behaviors through disinhibition of VTA GABA neurons that synapse onto VTA DA neurons and project back to the NAc[27]. NAc iSPNs first project to, and inhibit the ventral pallidum (VP); VP neurons then project to the VTA and ultimately the thalamus, resulting in inhibition of motivated behaviors[27].

With increasing use of cell type-specific approaches, it has become clear that the canonical divisions of the striatum: D1 vs D2 SPNs, dorsal vs ventral striatum, indirect vs direct pathway are over-simplified and there is likely further complexity within these circuits[28,29]. While it is well known that neuropeptides including opioids are active within the striatum, opioid receptors are G_{i/o}-coupled, and the opioids themselves are synthesized within the striatum[2]. In contrast, the *Grp* + DA neurons described above would bring in a neuropeptide from outside the striatum and are hypothesized to increase the excitability of GRPR-expressing neurons.

The goal of this study was to assess a possible role for GRP-GRPR signaling in the NAc. We find that *Grpr* is expressed in the NAc MSh, and *Grpr*-expressing cells have unique electrophysiological properties. Specifically, a majority of *Grpr*+ cells are spontaneously active in slice recordings and have high intrinsic excitability. Through fluorescent in situ hybridization (FISH) and analysis of scRNA-seq datasets[30], we identify *Grpr*+ cells as mainly D2R-expressing SPNs. Further, we find multiple distinct sub-populations of *Grpr*+ cells, both canonical iSPNs as well as putative eSPNs, based on gene expression patterns and electrophysiological signatures. In addition to the *Grp* + DA neurons, we show that the NAc MSh receives glutamatergic *Grp*+ inputs from the hippocampus and amygdala. GRP activates and increases the excitability of GRPR neurons in the NAc MSh both in slice and in vivo. Finally, we demonstrate that specific deletion of *Grpr* in the NAc MSh increases motivation in a progressive ratio test. These findings identify previously uncharacterized cell types in the NAc MSh and map out a GRP-GRPR circuit that mediates striatal excitability and motivated behaviors.

## Results

### Grpr is expressed in striatal neurons

To investigate whether GRP may be a functionally relevant neuromodulator in DA circuits, we used FISH to probe for expression of mRNA encoding *Grpr* in the DS and NAc MSh (Fig. 1a). Indeed, we found *Grpr* expression in subpopulations of cells in both the DS and NAc MSh (Fig. 1a). These cells were concentrated in the anterior portion of the striatum, with the highest concentration of *Grpr*+ cells found around A/P + 1.18 mm from bregma (Fig. 1b). Since *Grpr* is on the X chromosome, we analyzed *Grpr* expression by sex and found no difference in *Grpr* expression per cell or in the total number of *Grpr*+ cells in the NAc MSh between males and females (Supplementary Fig. 1a, b). Our findings are consistent with prior autoradiographic studies showing a high density of binding sites for bombesin-like peptides in the NAc and striatum[31,32]. Given the fact that the NAc was previously identified as a region with high GRPR expression[31,32], we focused further analysis on the *Grpr*-expressing cell population in the NAc MSh.

To study the properties of GRPR-expressing cells in the NAc MSh, we obtained a transgenic mouse line that fluorescently labels GRPR+ cells with eGFP (Fig. 1c–e). *Grpr*-eGFP mice (Gensat [MMRRC #036178-UCD])[33] have been shown to express eGFP exclusively in *Grpr*+ cells[9]. To characterize the morphology of GRPR+ cells in the NAc MSh, we filled cells with neurobiotin through a patch pipette and performed cellular reconstructions of eGFP+ and eGFP- cells (Supplementary Fig. 2a, b). We found that GRPR+ cells had similar dendritic complexity as neighboring cells (Supplementary Fig. 2c), the majority of which are expected to be SPNs. Additionally, we observed no differences in the total dendritic arbor length or the number of primary dendrites in eGFP+ vs eGFP- cells (Supplementary Fig. 2d, e). We also examined dendritic spines in cells transduced with an mCherry-expressing virus and observed that GRPR+ cells were spiny and had a similar spine density to neighboring neurons (Supplementary Fig. 2f–h). This analysis suggested that GRPR+ cells in the NAc MSh were likely a subpopulation of SPNs, as striatal interneurons have few or no spines[34].

We bred the *Grpr*-eGFP mice to *Drd1*-tdTomato mice[35] to determine whether the GRPR+ cells belonged to a given SPN subtype (Supplementary Fig. 2i, j). We noted that in the NAc MSh, the eGFP+ cells had little overlap with tdTomato+ cells (~5%) (Supplementary Fig. 2k). This, along with the presence of dendritic spines, indicated that the GRPR+ cells in the NAc MSh were likely a subpopulation of D2-expressing SPNs. As further confirmation of this, we bred *Grpr*-eGFP mice to *Adora2a*-Cre[36] mice together with a tdTomato Cre reporter line (Ai9)[37]. We found that ~93% of NAc MSh GFP+ cells expressed tdTomato in these mice (Supplementary Fig. 2l–n). Together, this suggested that *Grpr*-eGFP+ cells were likely iSPNs.

We next assessed the functional and physiological properties of GRPR-expressing neurons using the *Grpr*-eGFP;*Drd1*-tdTomato mice, which allowed us to visualize both GRPR+ cells and dSPNs. We used whole-cell electrophysiology to record from *Grpr*-eGFP+ cells and compared them to neighboring *Grpr*-eGFP- neurons in the NAc MSh (Fig. 1f, g). We also recorded from genetically identified dSPNs (D1-tdTomato + , Fig. 1h) and presumed iSPNs that were GRPR negative (D1-tdTomato-;*Grpr*-eGFP-, Fig. 1i). Interestingly, we found that the baseline physiological properties of *Grpr*-eGFP+ cells differed significantly from other SPNs in the NAc MSh. Approximately 75% of the *Grpr*-eGFP+ cells were spontaneously active at baseline (Fig. 1f), which was not characteristic of eGFP- cells or known SPN subtypes (Fig. 1g–i). Compared to neighboring cell populations, *Grpr*-eGFP+ neurons had a significantly more depolarized resting membrane potential (Fig. 1j) and higher membrane resistance (Fig. 1k). *Grpr*-eGFP+ cells differed in their input-output relationship compared to neighboring SPNs, exhibiting increased action potential firing in response to small positive currents and a transition to depolarization block with larger currents (Fig. 1l and Supplementary Fig. 3). *Grpr*-eGFP+ cells also differed in the shape of

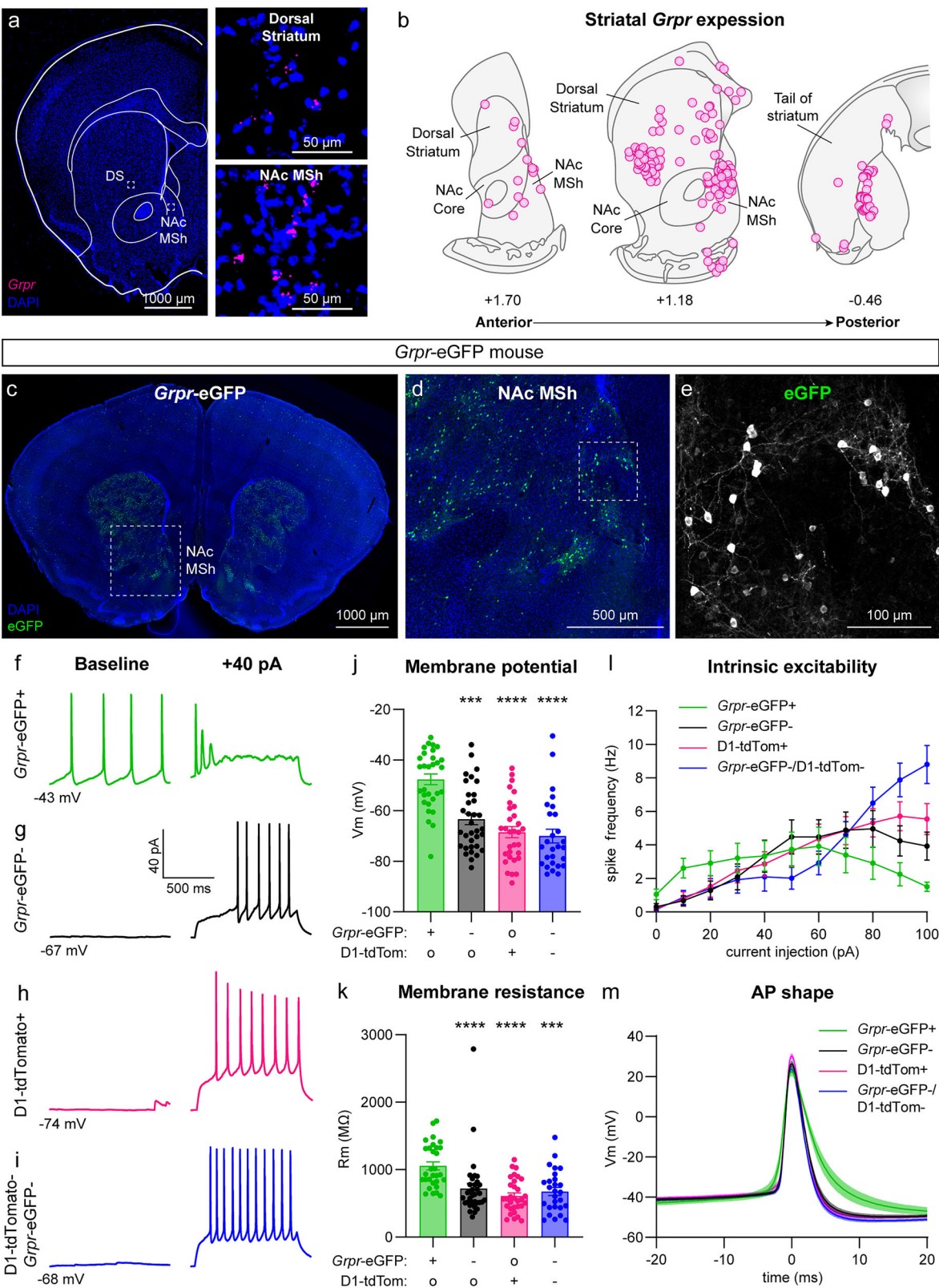

their action potentials (AP), particularly in terms of half-width (*Grpr*-eGFP+ 5.2970 ± 0.4821 ms vs. Grpr-eGFP- 3.2027 ± 0.1830 ms vs. D1-tdTomato+ 3.0003 ± 0.1076 ms vs. *Grpr*-eGFP-/D1-tdTomato- 2.9597 ± 0.1365 ms; ****p < 0.0001, one-way ANOVA; Holm-Sidak's multiple comparison tests, ****p < 0.0001 *Grpr*-eGFP+ vs *Grpr*-eGFP-, ****p < 0.0001 *Grpr*-eGFP+ vs D1-tdTomato + , ****p < 0.0001 *Grpr*-eGFP+ vs D1-tdTomato-/*Grpr*-eGFP-, Fig. 1m).

To determine whether expression of GRPR contributed to the high baseline excitability of *Grpr*-eGFP+ cells, we compared the resting membrane potential and membrane resistance of *Grpr*-eGFP+ cells with and without (D-Phe[6],Leu-NHEt[13], des-Met[14])-Bombesin (6-14) (DPDMB), a GRPR-specific antagonist. We found a decrease in the membrane resistance of *Grpr*-eGFP+ cells treated with DPDMB compared to untreated cells (Supplementary Fig. 4). This suggests that

**Fig. 1 | *Grpr* is expressed in the dorsal and ventral striatum and *Grpr*+ cells in the NAc MSh have unique physiological properties. a** Representative image of a coronal brain section from a wild-type (WT) mouse (representative of 3 mice). Fluorescent in situ hybridization (FISH) for *Grpr* in magenta. Nuclei are labeled in blue with DAPI. Insets show higher magnification images for dorsal striatum (DS) and nucleus accumbens medial shell (NAc MSh). **b** Summary schematic of *Grpr* expression across the A/P axis of the DS and NAc MSh. Each circle represents the location of a *Grpr*+ cell. Cells are summed across 3 WT mice. **c** Representative image of a coronal brain section from a *Grpr*-eGFP mouse (representative of 6 mice). eGFP is in green, DAPI labeled nuclei are in blue. **d** Zoom-in of the NAc MSh from panel (**c**). **e** Zoom-in of panel d with eGFP in grayscale. **f–i** Representative traces from *Grpr*-eGFP+ (f, representative of 23 neurons), *Grpr*-eGFP- (**g**, representative of 21 neurons), D1-tdTomato+ (**h**, representative of 31 neurons), and D1-tdTomato-/*Grpr*-

eGFP- (**i**, representative of 27 neurons) cells in the NAc MSh recorded at baseline (left) and following +40 pA current injection (right). **j** Mean ± SEM resting membrane potential of NAc MSh neurons. Dots represent values for individual cells. **k** Mean ± SEM membrane resistance of NAc MSh neurons. **l** Input-output curves showing the mean ± SEM firing frequency of NAc MSh neurons in response to positive current steps. **m** Mean ± SEM membrane potential for the first action potential (AP) evoked by current injection in each cell. Solid lines are the mean, shading represents SEM. For panels **j** and **k**, "+" indicates the presence of a fluorophore labeling the given cell type, "-" indicates that the cell was negative for the fluorophore, and "o" indicates that the tissue did not have a fluorophore that would label the given cell type. Source data are provided as a Source Data file. See the Supplementary Data file for sample sizes and statistics.

GRPR either has basal activity that can be blocked with DPDMB or that there may be tonic GRP-GRPR signaling in the slice preparation that increases the excitability of *Grpr*-eGFP+ cells.

Given that *Grpr*-eGFP+ cells were spontaneously active and had properties that were distinct from other SPNs in the NAc, we assessed how they compared to cholinergic interneurons, which are known to be tonically active[38] (Supplementary Fig. 5). We recorded from tdTomato-labeled neurons in *Chat*-Cre; Ai9 mice[37,39] and found that while the *Chat*-Cre+ cells tended to be spontaneously active (53% of cells were firing at baseline, Supplementary Fig. 5b), their resting membrane potential and membrane resistance were significantly different from *Grpr*-eGFP+ cells (Supplementary Fig. 5c, d). Additionally, with increasing positive current injections, the *Chat*+ cells continued to fire action potentials, while the *Grpr*-eGFP+ cells went into depolarization block with >60 pA current injection (Supplementary Fig. 5e). This, together with the fact that ChAT+ cells are known to be aspiny[40–42], indicated that GRPR+ cells are likely not a type of cholinergic interneuron.

## Grpr-expressing neurons in the NAc MSh are SPNs

Since *Grpr*-expressing cells had non-canonical properties compared to other neurons in the NAc MSh, we used multi-plex FISH with markers for known striatal cell types to determine their identity (Fig. 2a–j). We found that the vast majority of *Grpr*+ cells in the NAc MSh were positive for *Gad2* mRNA (96.3%) (Fig. 2a, b), and no *Grpr*+ cells were *Chat*+ (Fig. 2c, f), indicating that *Grpr*+ cells were GABAergic and not cholinergic. In the DS, we found similar expression patterns with 92.7% of *Grpr*+ cells expressing *Gad2*+ (Supplementary Fig. 6a, b) compared to only 1.9% co-expressing *Chat* mRNA (Supplementary Fig. 6c, f). To determine whether the *Grpr*+ cells were SPNs or a type of GABAergic interneuron, we probed for *Ppp1r1b*, which encodes the SPN-marker DARPP-32 (Fig. 2c, g). Nearly all *Grpr*-expressing cells co-expressed *Ppp1r1b* in the NAc MSh (93.8%) (Fig. 2g) and DS (75.5%) (Supplementary Fig. 6c, g), indicating that they were SPNs.

To determine the specific SPN identity of the *Grpr*+ cells, we probed for mRNA encoding D1 (*Drd1*) and D2 (*Drd2*) DA receptors (Fig. 2d), which are expressed in dSPNs and iSPNs, respectively. The majority of *Grpr*+ cells in the NAc MSh expressed *Drd2* mRNA (77.3%) (Fig. 2h), suggesting that they were iSPNs. However, a small subpopulation of *Grpr*+ cells in the NAc MSh expressed *Drd1* (26.7%) (Fig. 2i). Multiple studies have reported that a proportion (5-30%) of SPNs in the NAc MSh express both *Drd1* and *Drd2*[28,43,44]. Therefore, we examined whether the *Grpr*+/*Drd1*+ population in the NAc MSh were dSPNs or rather SPNs co-expressing both DA receptors. We found that 61.5% of *Grpr*+ cells in the NAc MSh expressed *Drd2*, but not *Drd1*, 14.1% expressed *Drd1* only, and 12.6% expressed both *Drd1* and *Drd2* (Fig. 2k). Given the preferential expression of *Drd2* in *Grpr*+ cells, we probed for the iSPN marker *Penk* (Fig. 2e). We found that it was expressed by the majority (67.4%) of *Grpr*+ cells in the NAc MSh (Fig. 2j). We also examined the co-localization between *Drd2* and *Penk* mRNA and found that 22.2% of the *Grpr*+ cells in the NAc MSh expressed *Drd2* but not

*Penk*, 4.3% expressed *Penk* but not *Drd2*, and 63.1% of *Grpr*+ cells in the NAc MSh expressed both markers (Fig. 2l). Together, this analysis indicates that *Grpr*+ cells in the NAc MSh are likely to be GABAergic D2R-expressing SPNs. We note that only a small percentage of all *Ppp1r1b* (1.5%), *Drd2* (3.9%), or *Penk*-expressing (2.8%) cells in the MSh were *Grpr* positive, indicating that *Grpr*+ cells are a select subpopulation of SPNs in the NAc MSh (Fig. 2m–o).

We next examined the SPN identity of *Grpr*+ cells in the DS. While a majority of *Grpr*+ cells in the DS expressed SPN markers (Supplementary Fig. 6c–j), the SPN sub-type classification was less clear. *Drd2* mRNA was expressed in 32.7% of DS *Grpr*+ cells, while *Drd1* mRNA was expressed in 43.3% of *Grpr*+ cells (Supplementary Fig. 6h, i). We also found that *Penk* mRNA was found in 38.0% of DS *Grpr*+ cells, consistent with the proportion expressing *Drd2* mRNA (Supplementary Fig. 6j). Together this suggests that *Grpr* is expressed in both dSPNs and iSPNs in the DS.

We were intrigued by the population of *Grpr*+ cells in the NAc MSh that expressed mRNA for *Drd2*, but not the other canonical iSPN marker, *Penk*. We therefore made use of the whole mouse brain scRNA-seq atlas from the Allen Institute for Brain Science[30] to further explore the molecular identity of *Grpr*+ cells (Supplementary Fig. 7). We first extracted and clustered all the cells in the mouse dorsal and ventral striatum (Supplementary Fig. 7a, b). We noted a clear separation between the DS and VS within the largest SPN clusters, defined by *Ppp1r1b* expression (Supplementary Fig. 7b, c). Feature plots highlighting *Ppp1r1b*, *Drd1*, *Drd2*, and *Grpr*-expressing cells were consistent with FISH results showing *Grpr*+ cells were mainly *Ppp1r1b* + /*Drd2*+ in the VS (Supplementary Fig. 7c–f).

We then subsetted the *Grpr*+ cells in the VS and clustered them into five distinct clusters (Supplementary Fig. 7g). The two largest clusters of *Grpr*-expressing cells (clusters 0 and 1) expressed *Gad2*, *Ppp1r1b*, and *Drd2*, but not *Drd1*, *Chat*, *Pvalb*, or *Sst* (Supplementary Fig. 7h–p), consistent with an iSPN identify. We noted that cluster 1 cells expressed *Drd2* but had lower expression of *Penk* (Supplementary Fig. 7l, m), reminiscent of the *Grpr* + /*Drd2* + /*Penk*- cells observed by FISH. We further explored the identity of cluster 1 cells and found that they expressed markers of eSPNs[21] including *Casz1*, *Th*, *Otof*, *Cacng5*, and *Pcdh8* (Supplementary Fig. 7q-u). Together, this suggests that the majority of *Grpr*+ cells in the VS are GABAergic *Drd2*-expressing SPNs that can be further divided into iSPN and eSPN sub-types based on their gene expression.

To test whether *GRPR*-expressing cells are also present in the human caudate/putamen, we applied a similar analysis to a scRNA-seq dataset from the human adult brain[45] (Supplementary Fig. 8). Feature plots of *PPP1R1B*, *DRD1*, *DRD2*, and *GRPR* expression showed that in the human VS, *GRPR* is expressed in both *DRD1*- and *DRD2*-expressing SPNs (Supplementary Fig. 8b–e). We isolated the *GRPR*-expressing cells in the VS and clustered them into four clusters (Supplementary Fig. 8f). In the human brain dataset, clusters 0, 1, and 2 expressed SPN markers, with cluster 0 corresponding to *DRD1*-expressing dSPNs, cluster 1 to *DRD2*-expressing iSPNs, and cluster 2 to *CASZ1*-expressing eSPNs (Supplementary Fig. 8f–t). Interestingly, unlike in the mouse dataset, there was also a population of *GRPR* + *SST*-expressing cells, which may

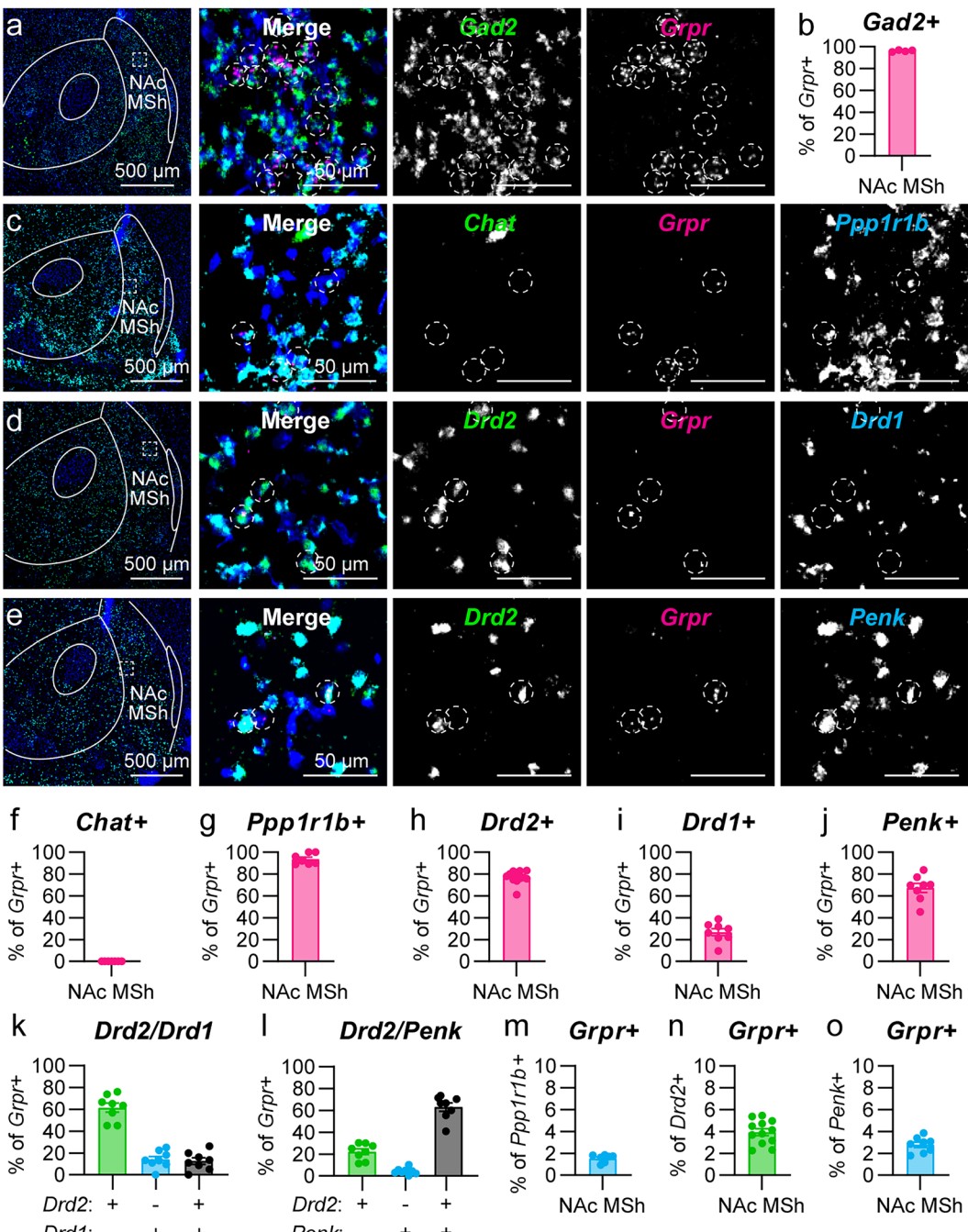

**Fig. 2 | *Grpr* is expressed in SPNs in the NAc MSh. a** Representative image of the NAc MSh (representative of 4 mice). FISH for *Gad2* in green and *Grpr* in magenta. Nuclei labeled with DAPI in blue. Right panels are zoomed-in images from the boxed region. Dashed circles outline *Grpr*+ cells. **b** Mean ± SEM percentage of *Grpr*+ cells in the NAc MSh and that are *Gad2*+ in WT mice. (*n* = 4 mice). **c** Representative FISH image with *Chat* in green, *Grpr* in magenta, and *Ppp1r1b* in cyan (representative of 7 mice). **d** FISH for *Drd2* in green, *Grpr* in magenta, and *Drd1* in cyan (representative of 8 mice). **e** FISH for *Drd2* in green, *Grpr* in magenta, and *Penk* in cyan (representative of 8 mice). (**f**–**j**) Mea*n* ± SEM percentage of *Grpr*+ cells in the NAc MSh of WT mice that are **f** *Chat* + (*n* = 7 mice), **g** *Ppp1r1b* + (*n* = 7 mice), **h** *Drd2* + (*n* = 12 mice), **i** *Drd1* + (*n* = 8 mice), or **j** *Penk* + (*n* = 8 mice). **k** Mean ± SEM

percentage of *Grpr*+ cells in the NAc MSh that are *Drd2* + /*Drd1*- (green), *Drd2*-/*Drd1* + (blue), or *Drd2* + /*Drd1* + (black), *n* = 8 WT mice). **l** Mean ± SEM percentage of *Grpr*+ cells in the NAc MSh that are *Drd2* + /*Penk*- (green), *Drd2*-/*Penk* + (blue), or *Drd2* + /*Penk* + (black). (*n* = 8 WT mice). **m**–**o** Mean ± SEM percentage of **m** *Ppp1r1b* + (*n* = 7 mice), **n** *Drd2* + (*n* = 12 mice), or **o** *Penk*+ cells (*n* = 8 mice) in the NAc MSh that are *Grpr* + . For all bar graphs, dots represent values for individual mice. Values in panels **b**, **f**–**o** were obtained by counting all cells in the NAc MSh from two independent slices per mouse that were summed together. Example images show only a portion of the NAc MSh. Source data are provided as a Source Data file.

represent somatostatin interneurons (cluster 3, Supplementary Fig. 8o). Overall, while there was more diversity in the types of cells that expressed *GRPR* in the human VS, it was clear that *GRPR* is expressed in subpopulations of iSPNs and eSPNs, consistent with our observations in mice.

## GRPR+ cells in the NAc MSh comprise distinct physiological sub-types

Given that the scRNA-seq analysis had identified a population of *Grpr*+ cells in the VS that appeared to be eSPNs, we completed additional FISH experiments to probe for the eSPN marker *Casz1* in the NAc MSh

(Fig. 3a–c). We found that 36.4% of *Grpr*+ cells in the NAc MSh expressed *Casz1* mRNA (Fig. 3b). This was notable given that 63.1% of *Grpr*+ cells in the NAc MSh expressed both *Drd2* and *Penk* (Fig. 2l); therefore, *Grpr + /Casz1*+ cells could represent the remaining fraction. This provides further evidence that there are at least two populations of *Grpr*+ cells in the NAc MSh: classical iSPNs and eSPNs. We also noted that not all eSPNs were *Grpr* + , as only 15.6% of *Casz1*+ cells in the NAc MSh were *Grpr* + (Fig. 3c).

Given this insight, we generated pair plots to examine how passive properties were distributed among GRPR+ SPNs and genetically defined dSPNs (D1), and iSPNs (D2) (see Fig. 1). We found that for both resting membrane potential and membrane resistance, the distribution of values for GRPR+ cells was shifted to the right compared to other SPNs and there were two distinct peaks for the GRPR+ population (Fig. 3d). The distributions for membrane capacitance were overlapping among all cell types, consistent with the similar morphological profile of GRPR+ and GRPR- SPNs in the NAc MSh (see Supplementary Fig. 2). We next performed dimensionality reduction and clustering analysis on the physiology data from Fig. 1 (Fig. 3e, f). We used information about each cell's resting membrane potential, membrane resistance, and capacitance to generate a 2D UMAP embedding of the cells. Informed by this UMAP, we clustered the cells into 4 distinct clusters (Fig. 3e). GRPR+ cells belonged to three of these clusters: 0, 1, and 3 (Fig. 3f), indicating multiple distinct populations.

To delve into this further, we separated the GRPR+ cells based on their cluster identity and compared them to known D1+ and presumed D2+ cells in the NAc MSh (Fig. 3g–l). We examined resting membrane potential, membrane resistance, capacitance, AP half-width, and AP height (Fig. 3g–k). We found no significant differences in any of the examined properties between the SPN groups and the GRPR+ cells in cluster 1 (Fig. 3l). However, GRPR+ cells in clusters 0 and 3 had more depolarized resting membrane potential and greater AP half-width compared to canonical SPNs (Fig. 3h, k). Notably, GRPR+ cells in cluster 3 also had significantly higher membrane resistance than the SPN groups (Fig. 3h).

In both the *Casz1* FISH experiment and in the *Grpr*-eGFP mice, we noted that there were instances of high-density anatomical clusters of cells in the ventral portions of the NAc MSh (Supplementary Fig. 9a–c). In the FISH experiments, *Grpr* and *Casz1* mRNA were highly co-expressed in these regions (Supplementary Fig. 9a). These clusters were reminiscent of immature cells migrating towards their final adult location[46]. Indeed, it was recently shown by scRNA-seq that *Casz1* is expressed in immature SPNs in the mouse striatum at P8[47]. We therefore hypothesized that some of the GRPR+ cells could be immature SPNs. To test this, we recorded from iSPNs in the NAc MSh from juvenile postnatal day p 8-12 D2-GFP mice (Supplementary Fig. 9d–i). Cluster 0 and 1 GRPR+ cells had lower membrane resistance and higher capacitance compared to the immature cells, consistent with changes that occur in SPNs during postnatal maturation[48,49]. However, cluster 3 cells were not significantly different from the immature cells across any of the measures (Supplementary Fig. 9d–i). This analysis shows that cluster 1 GRPR+ cells are most similar in their properties to mature canonical SPNs, while cluster 3 GRPR+ cells may represent iSPNs that are in an immature state. We hypothesize that cluster 0 GRPR+ cells, which were distinct from both mature and immature SPNs, may be eSPNs. However, since the physiological properties of eSPNs have yet to be defined, this will require further investigation. Together, these analyses reveal multiple sub-populations of NAc GRPR+ neurons defined by their physiological properties.

## GRPR+ neurons in the NAc MSh are more likely to be Fos+ in vivo
Given the high excitability of *Grpr*-eGFP+ cells in our slice recordings, we examined whether they may be more active in vivo. To assess this, we performed FISH in naïve wild-type (WT) mice and probed for *Drd2, Grpr*, and *Fos* mRNA. We assessed the percentage of *Drd2*+ cells in the NAc MSh that were *Fos*+ under baseline conditions, using *Fos* as a proxy for neuronal activity. We found that *Drd2 + /Grpr*+ cells were over twice as likely to be *Fos*+ compared to neighboring *Drd2 + /Grpr*- cells (18.7% compared to 8.6% *Fos* + ) (Fig. 4a, g). Since *Grpr* is expressed in both iSPNs and eSPNs, we repeated the experiment and probed for *Penk* to define iSPNs (Fig. 4b, h) or *Casz1* to label eSPNs (Fig. 4c, i). In both iSPNs and eSPNs we saw similar results, whereby the *Grpr*-expressing subpopulation was significantly more likely to express *Fos* compared with neighboring *Grpr*-cells (Fig. 4h, i). We note that the total *Grpr*+ population represented only a fraction (<7%) of total *Fos*+ cells in the NAc MSh (Fig. 4j).

To test whether the increased excitability seen both in slice and in vivo was due to GRPR activation or some other mechanism unique to *Grpr*-eGFP+ cells, we generated *Grpr* knockout (KO) mice. To do this, we obtained floxed *Grpr* (*Grpr^fl*) mice, which contain loxP sites flanking exon 2[11], and bred them to CMV-Cre mice[50] to induce constitutive deletion of *Grpr* (Supplementary Fig. 10a–d). We verified that the *Grpr* FISH probe binding site was not located within exon 2 and could be used to identify cells that normally expressed *Grpr*, even if the gene was no longer functional (we term cells that typically express *Grpr* as "*Grpr*-lineage cells"). We first examined whether there was any difference in the percentage of *Grpr*-lineage cells that were *Fos*+ between WT and *Grpr* KO mice (Supplementary Fig. 10e–h). We found that disrupting *Grpr* expression significantly reduced the percentage of *Grpr*-lineage cells that were *Fos*+ in naïve mice without affecting the total number of *Fos*+ cells in the NAc MSh (Supplementary Fig. 10e–h).

We next determined the effect of *Grpr* deletion on *Fos* expression across different SPN subtypes. *Grpr* KO mice did not show a significant difference in the percentage of *Fos*+ cells between *Drd2*+ or *Penk*+ *Grpr*-lineage cells and *Grpr*- cells (Fig. 4d, e, g, h). This suggests that expression of *Grpr* is responsible for the increased activation of these cells. Interestingly, when we examined eSPNs, we found that *Casz1 + /Grpr*-lineage cells were still more likely to be *Fos*+ compared to *Casz1 + /Grpr*- cells (Fig. 4f, i). Thus, *Grpr* expression is required for the increased activation state of *Grpr*-expressing iSPNs, but not eSPNs.

## Multiple inputs to the NAc MSh express Grp mRNA
While it was previously established that *Grp + DA* neurons project to the NAc MSh from the VTA[5], the NAc MSh receives inputs from a number of other brain regions including the hippocampus, amygdala, thalamus, entorhinal cortex, and others[51]. Although GRP and GRPR are relatively sparsely expressed in the brain, several of these brain areas have been reported to express the mRNA for *Grp*[52]. Therefore, we asked whether other inputs into the NAc MSh might have the potential to co-release GRP. To investigate this, we injected retrobeads into the NAc MSh and performed FISH for *Grp* in the ten primary brain regions that send projections to the NAc MSh[51] (Fig. 5a–f and Supplementary Fig. 11). We found a high percentage of retrobead+ cells that were also *Grp*+ in the VTA, ventral hippocampus (subiculum and CA1), and basolateral amygdala (BLA) (Fig. 5b–f). We found little to no expression of *Grp* in other regions projecting to the NAc MSh, including the lateral entorhinal cortex, anterior piriform cortex, and thalamus, amongst others (Fig. 5b). This suggests that a select group of afferents to the NAc MSh could have the capacity to co-release GRP.

We summed together all the *Grp*+ inputs quantified and found that the main sources of *Grp* into the NAc MSh were from the subiculum (54%), CA1 (31%), VTA (6%), and BLA (5%) (Fig. 5g, h). We further analyzed the CA1, subiculum, and BLA for *Slc17a7* (excitatory neuron marker) and *Gad2* (inhibitory neuron marker) expression and found that nearly all the *Grp* inputs from these regions were glutamatergic (Fig. 5i-k). This suggests that in addition to dopaminergic inputs,

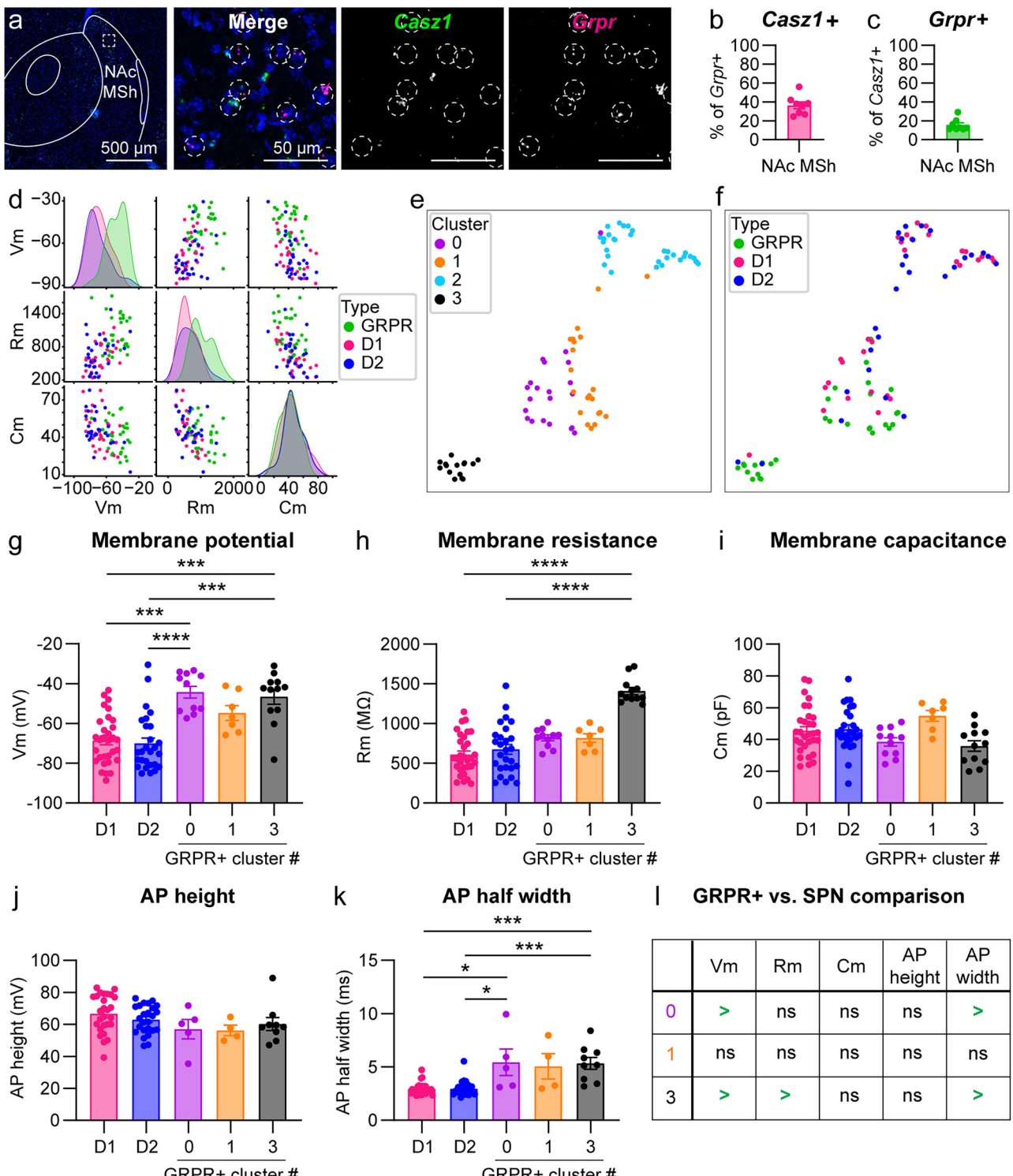

**Fig. 3 | Subpopulations of *Grpr+* neurons in the NAc MSh have distinct physiological properties. a** Representative image of the NAc MSh (representative of 8 mice). FISH for *Casz1* in green and *Grpr* in magenta. Nuclei are labeled with DAPI in blue. Dashed circles outline *Grpr+* cells. **b** Mean ± SEM percentage of *Grpr+* cells in the NAc MSh and that are *Casz1 +* (*n* = 8 WT mice). **c** Mean ± SEM percentage of *Casz1+* in the NAc MSh that are *Grpr +* (*n* = 8 WT mice). For **b** and **c**, dots represent values for individual mice. Values were obtained by counting all cells in the NAc MSh from two independent slices per mouse that were summed together. **d** Pair plot of passive properties from electrophysiology recordings from D1 (D1-tdTomato + ), D2 (D1-tdTomato-/*Grpr*-eGFP-), and Grpr (*Grpr*-eGFP + ) cells in the NAc MSh. Dots represent values for individual cells. Shaded regions represent the distribution of values for the indicated cell types. **e** UMAP of passive properties of D1,

D2, and GRPR+ cells in the NAc MSh. Vm, Rm, and Cm values for each cell were centered and scaled to unit variance, and UMAP embeddings were computed using n_neighbors=10 and min_dist=0. Clusters were determined using the K-Means algorithm with n_clusters=4. **f** UMAP of passive properties from panel e colored by cell type. **g** Mean ± SEM resting membrane potential of NAc MSh neurons from the indicated groups. **h** Mean ± SEM membrane resistance. **i** Mean ± SEM membrane capacitance. **j** Mean ± SEM action potential (AP) height. **k** Mean ± SEM AP half-width. For **g**–**k**, dots represent values for individual cells. **l** Summary table of **g**–**k**. Data for D1+ and D2+ cells in g-i is the same as in Fig. 1, plotted here for comparison. Data for GRPR+ cells in **g**–**i** is the same as in Fig. 1 but subdivided by cluster. Source data are provided as a Source Data file. See the Supplementary Data file for sample sizes and statistics.

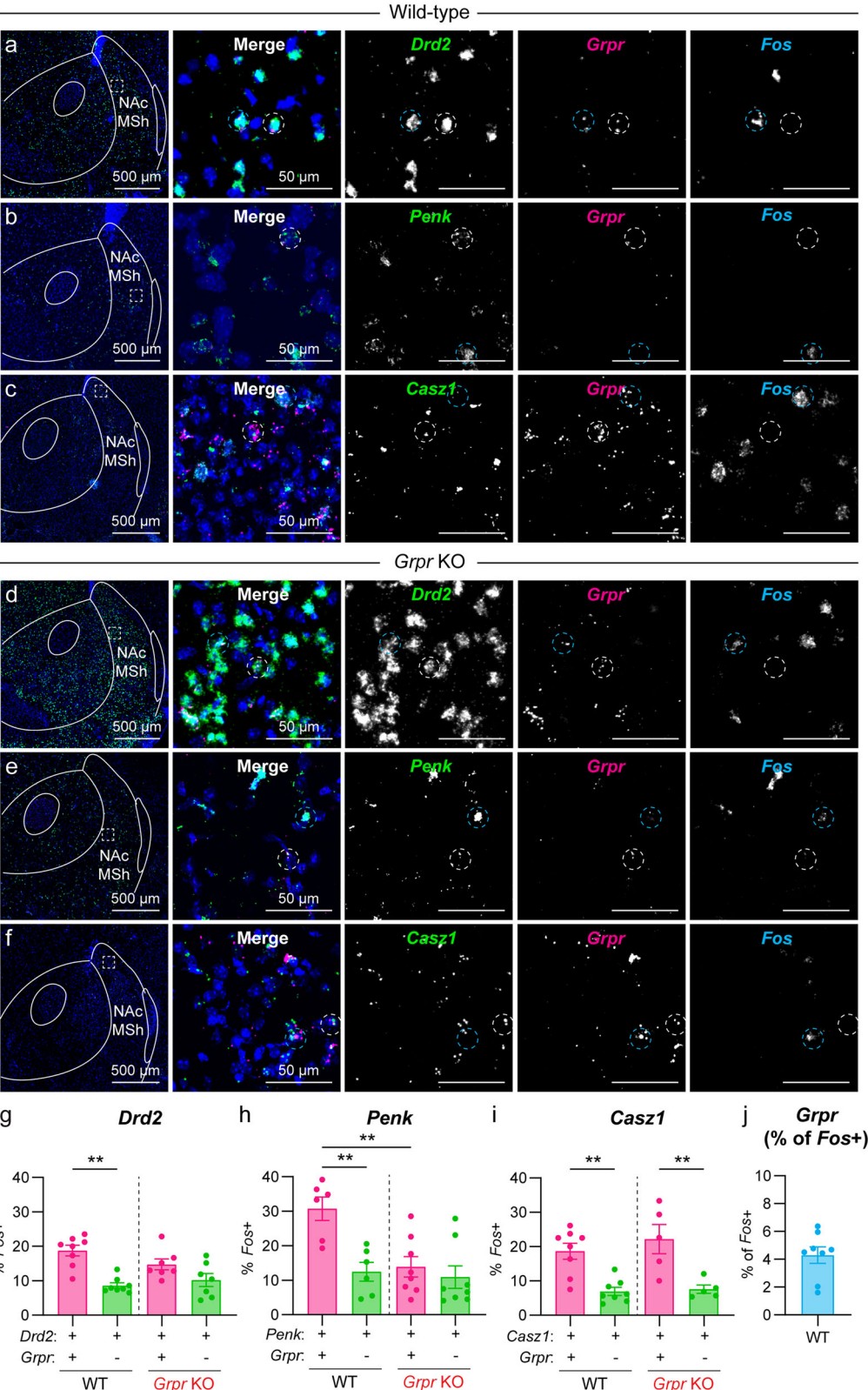

several different glutamatergic inputs to the NAc MSh may have the ability to use GRP as a neuromodulator.

## GRP depolarizes and increases the excitability of GRPR-expressing neurons

After establishing that GRPR+ cells in the NAc MSh have distinct electrophysiological properties and mapping the *Grp*+ inputs to this region, we sought to determine how GRP affects the functional properties of these cells. To do this, we washed-on GRP to NAc MSh slices and measured resting membrane potential and membrane resistance in *Grpr*-eGFP+ cells. We found that 300 nM GRP significantly depolarized *Grpr*-eGFP+ cells and increased their membrane resistance (Fig. 6a, c, d), consistent with the effects of GRP in other cell types[9,53,54]. Importantly, these changes were blocked in the presence of the

**Fig. 4 | *Grpr*+ cells in the NAc MSh have increased *Fos* expression.**
**a** Representative image of the NAc MSh from a WT mouse (representative of 8 mice). FISH for *Drd2* in green, *Grpr* in magenta, and *Fos* in cyan. Right panels show zoomed-in images of the boxed region. Dashed circles outline sample *Grpr*+ cells, white=*Grpr* + /*Drd2* + /*Fos*-, cyan=*Grpr* + /*Drd2* + /*Fos* + . **b** FISH for *Penk* in green, *Grpr* in magenta, and *Fos* in cyan (representative of 6 mice). Dashed circles outline sample *Grpr*+ cells, white=*Grpr* + /*Penk* + /*Fos*-, cyan=*Grpr* + /*Penk* + /*Fos* + . **c** FISH for *Casz1* in green, *Grpr* in magenta, and *Fos* in cyan (representative of 8 mice). Dashed circles outline sample *Grpr*+ cells, white=*Grpr* + /*Casz1* + /*Fos*-, cyan=*Grpr* + /*Casz1* + /*Fos* + . **d**–**f** Same as **a**–**c**, but in *Grpr* knock-out (KO) mice. **d** is representative of 7 mice. **e** is representative of 8 mice. **f** is representative of 5 mice. **g** Mean ± SEM

percentage of *Drd2* + /*Grpr* + (magenta) vs. *Drd2* + /*Grpr*- (green) cells in the NAc MSh of WT mice (left) and *Grpr* KO mice (right) that are *Fos* + . **h** Mean ± SEM percentage of *Penk* + /*Grpr* + (magenta) vs. *Penk* + /*Grpr*- (green) cells in the NAc MSh of WT mice (left) and *Grpr* KO mice (right) that are *Fos* + . **i** Mean ± SEM percentage of *Casz1* + /*Grpr* + (magenta) vs. *Casz1* + /*Grpr*- (green) cells in the NAc MSh of WT mice (left) and *Grpr* KO mice (right) that are *Fos* + . **j** Mean ± SEM percentage of *Fos*+ in the NAc MSh that are *Grpr* + (n = 8 WT mice). For all bar graphs, dots represent values for individual mice. Values in panels **g**–**j** were obtained by counting all cells in the NAc MSh from two independent slices per mouse that were summed together. Source data are provided as a Source Data file. See the Supplementary Data file for sample sizes and statistics.

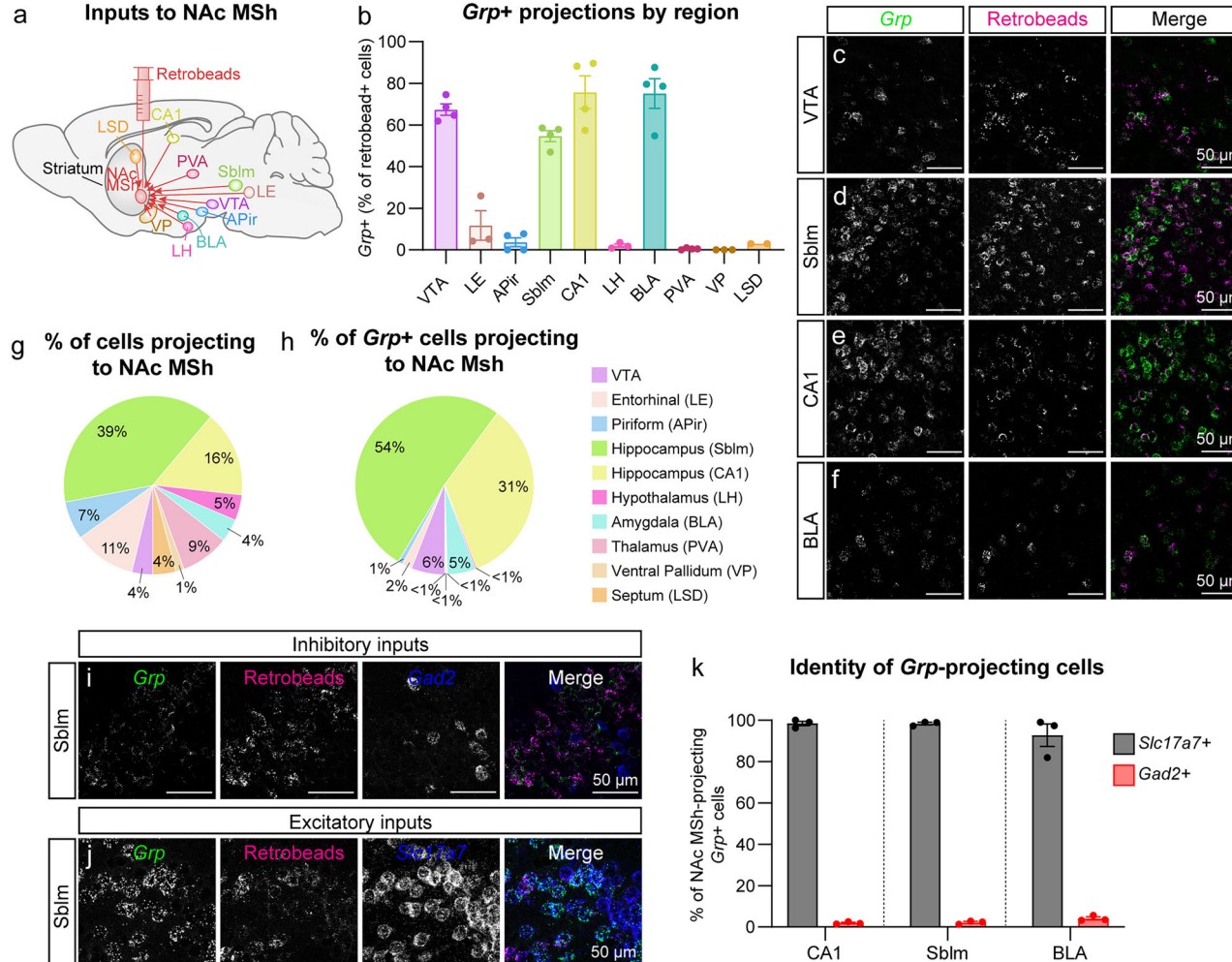

**Fig. 5 | NAc MSh-projecting glutamatergic neurons from the hippocampus and amygdala express *Grp*. a** Schematic of retrograde tracing experiment. Retrobeads were injected into the NAc MSh and the listed projection targets were examined for *Grp* mRNA and retrobead co-localization using FISH. NAc MSh inputs: ventral tegmental area (VTA), lateral entorhinal cortex (LE), anterior piriform (APir), hippocampus subiculum (Sblm), hippocampus CA1 (CA1), lateral hypothalamus (LH), basolateral amygdala (BLA), paraventricular thalamus (PVA), ventral pallidum (VP), and dorsal lateral septum (LSD). **b** Mean ± SEM percentage of total retrobead+ cells that were *Grp*+ in each of the examined regions (dots represent values for four independent injection sites from three mice). **c**–**f** Representative FISH images from the indicated brain regions showing *Grp*+ cells in green and retrobead+ cells in magenta: **c** VTA, **d** subiculum, **e** CA1, and **f** BLA. Each image is representative of four

independent injection sites from three mice. **g** Pie chart displaying the percentage of total retrobead+ cells coming from each region. Retrobead+ cells were summed across all mice (n = 4 injections from 3 mice) and then divided by the number of retrobead+ cells in each region. **h** Pie chart showing the contribution of each input region to the total population of *Grp*+ neurons projecting to the NAc MSh. **i**, **j** Representative FISH images of the subiculum showing *Grp*+ cells in green, retrobead+ cells in magenta, and **i** *Gad2*+ or **j** *Slc17a7*+ cells in blue (representative of three injection sites from three mice). **k** Mean ± SEM percentage of retrobead + /*Grp*+ cells that were *Gad2*+ or *Slc17a7*+ for the three main *Grp*+ inputs into the NAc MSh (dots represent values for three independent injection sites from three mice). Source data are provided as a Source Data file.

GRPR-specific antagonist, DPDMB (Fig. 6b, e, f). As a control, we performed the same experiment on D2-GFP+ cells in D2-GFP mice[33], the majority of which are not expected to express GRPR (see Fig. 2n). We

did not detect any change in membrane potential or resistance with GRP wash-on (Supplementary Fig. 12), demonstrating that GRP selectively increases the excitability of GRPR-expressing neurons.

## GRP injection in vivo increases Fos expression

We next asked whether GRP activates GRPR+ neurons in vivo and whether it has broader network effects in the NAc MSh. We injected 3 μM GRP unilaterally into the NAc MSh of anesthetized WT mice and harvested the brains 45 minutes later (Fig. 6g)[53]. We performed FISH for *Drd2*, *Grpr*, and *Fos* mRNA and compared the injected hemisphere to the un-injected hemisphere (Fig. 6h, i). We found an increase in the total number of *Fos*+ cells in the GRP-injected hemisphere compared to the control, indicating an increase in activated cells upon GRP injection (Fig. 6j). The total number of *Grpr*+ cells was unchanged, as expected (Fig. 6k). *Fos* was selectively induced in *Grpr*+ cells (Fig. 6l-n) as the percentage of *Drd2+/Grpr-* cells that were *Fos* positive was unchanged (Fig. 6o). This shows that GRP selectively activated GRPR-expressing cells and did not induce larger network-level activation. To confirm that *Fos* activation in *Grpr*+ cells was due to GRPR signaling, we injected GRP into the NAc of *Grpr* KO and littermate WT mice. We again observed an increase in the percentage of *Grpr*+ cells that were *Fos*+ in WT mice (Supplementary Fig. 13a–d), which was not observed in *Grpr* KO mice (Supplementary Fig. 13e–h). Given that *Fos* is a proxy for neuronal activity, these data indicate that GRP selectively increases the activity of *Grpr*+ cells in the NAc MSh via activation of GRPR.

As a control to verify that this effect was induced by GRP and not due to the injection itself, we repeated the above experiment in WT mice but injected sterile saline instead of GRP (Supplementary Fig. 14). While we saw a small increase in the total number of *Fos*+ cells in the saline-injected hemisphere, there was no change in the total number of *Grpr*+ cells or increase in the *Grpr*+ cells that were *Fos* + (Supplementary Fig. 14a–e). Additionally, we saw no increase in the percentage of *Grpr*+ cells that were *Fos* + (Supplementary Fig. 14f). Together, these results indicate that while saline injection can acutely increase neuronal activity in the NAc MSh, it does not preferentially activate *Grpr*+ cells.

## Deletion of Grpr in the NAc MSh selectively increases motivation

Given that GRP-GRPR signaling strongly modulated the excitability of a subset of NAc MSh neurons, we tested whether NAc MSh-specific deletion of *Grpr* affected striatal-dependent behaviors. To generate conditional *Grpr* knock-out (cKO) mice, we injected *Grpr*^fl/fl or *Grpr*^fl/y mice[11] bilaterally with an AAV expressing Cre recombinase, to specifically delete *Grpr* in the adult NAc MSh (Fig. 7a, b). A GFP-expressing virus was used as a control. Four weeks post-surgery, *Grpr* cKO mice and control GFP-injected littermates were run through the open field test, accelerating rotarod, progressive ratio (PR) lever-pressing task, and sucrose preference test to assess motor and reward-related behaviors (Supplementary Fig. 15a).

The open field assay was used to measure general locomotor behavior and was analyzed using DeepLabCut (Supplementary Fig. 15b)[55]. We found that *Grpr* cKO mice traveled a greater total distance during the one-hour session (Supplementary Fig. 15c), suggesting a mild increase in locomotor activity. We assessed the time spent in the center, as a proxy for avoidance behavior, and number of grooming bouts, as a measure of repetitive behavior, and found no differences in either (Supplementary Fig. 15d, e). We used Keypoint-MoSeq[56,57] to perform unbiased identification of spontaneous behavior syllables used by the mice in the open field. The behavior syllables detected by this approach included movements like forward running, right or left turn, and grooming (Supplementary Fig. 15f, g). We found no significant differences between groups in their usage of any of the top 28 most frequent syllables (Supplementary Fig. 15f). Together, this suggests that disruption of GRPR signaling in the NAc MSh does not have a major impact on general behavior patterns.

Both the DS and VS have been implicated in motor learning[58]. To examine whether *Grpr* deletion impacts motor coordination or motor routine learning, we tested mice on the accelerating rotarod with three trials per day across four consecutive testing days. We saw no

differences in motor coordination or learning between *Grpr* cKO mice and GFP-injected controls (Supplementary Fig. 16a–d).

The NAc is a central component of the mesolimbic reward processing pathway. Alterations in NAc activity can therefore affect the hedonic value of natural rewards[59,60]. To test whether *Grpr* signaling is involved in this, we compared *Grpr* cKO and control mice in the sucrose preference test (Supplementary Fig. 16e). This test is a measure of hedonia, or the ability to derive pleasure from rewarding stimuli[61–63]. We first verified there was no difference in starting weight between groups (Supplementary Fig. 16f). During the test sessions, we found that *Grpr* cKO and control mice showed a similar preference for the sucrose solution over plain water (Supplementary Fig. 16g). These results suggest that GRPR signaling in the NAc MSh is not required for reward value processing.

In addition to processing reward, the NAc is implicated in the control of motivated behaviors. In particular, modulation of D2 SPNs in the NAc MSh has been shown to impact motivation[64,65]. For example, chemogenetic inhibition of the indirect pathway NAc outputs to the VP is sufficient to enhance motivation[64], suggesting an inhibitory role for this pathway in motivated behavior. To examine how GRPR signaling impacts motivation, we tested *Grpr* cKO mice on a PR lever pressing task (Fig. 7c). Prior to training, mice were food restricted to 90-95% of their body weight. The mice were trained to press a lever for a food pellet reward in daily sessions with a fixed-ratio of 1 (one press yields one reward, FR1) until they successfully received 20 or more rewards in a one-hour session on two consecutive days. No differences were seen in the number of days it took mice to learn the task (Fig. 7d). There were also no genotype differences in lever-pressing behavior across FR1 training days (Supplementary Fig. 17a, b). Additionally, there was no difference in the starting weight of mice or the percentage of their initial body weight on the PR day between groups (Fig. 7e, f). During the two-hour PR session, mice had to press the lever an increasing number of times to earn food pellet rewards. The PR session was considered over after three minutes had elapsed without a lever press, after the mouse had engaged with the lever at least once. Survival function analysis revealed a significant difference in the number of mice that continued to engage in the task between groups, with more *Grpr* cKO mice demonstrating prolonged pressing (Fig. 7g). Additionally, *Grpr* deletion in the NAc MSh led to a significant increase in the total number of lever presses during the PR session (Fig. 7h), as well as the breakpoint, which is the maximum number of lever presses a mouse is willing to perform for a single pellet reward (Fig. 7i). *Grpr* cKO mice also showed a significant increase in the total PR session time (Fig. 7j, k). These results indicate that disruption of GRP-GRPR signaling in the NAc MSh enhances motivation.

## Discussion

Neuropeptides are potent neuromodulators that regulate neuronal activity and communication in a variety of ways[66,67]. It has recently become appreciated that many neurons co-express genes for both classical neurotransmitters and one or more neuropeptides[4,5]. As such, neuropeptides are poised to be key regulators of neural circuit function. Here we identified a role for the neuropeptide GRP and its receptor GRPR in mediating the excitability and activity of neurons in the NAc MSh. Specifically, we find that *Grpr*+ neurons in the NAc MSh are a subpopulation of D2R-expressing SPNs with distinct electrophysiological properties. The *Grpr*+ cells can be further subdivided into traditional iSPNs as well as eSPNs based on their genetic markers. We mapped the *Grp*-expressing inputs into the NAc MSh and showed that in addition to the previously identified DA neurons from the VTA, the NAc MSh receives significant *Grp*+ inputs from glutamatergic neurons in the ventral hippocampus and amygdala. Furthermore, we show both in slice and in vivo that the *Grpr*+ neurons in the NAc MSh have high basal excitability and can be further excited by GRP. Finally, we eliminate NAc GRP-GRPR signaling by generating *Grpr* cKO mice

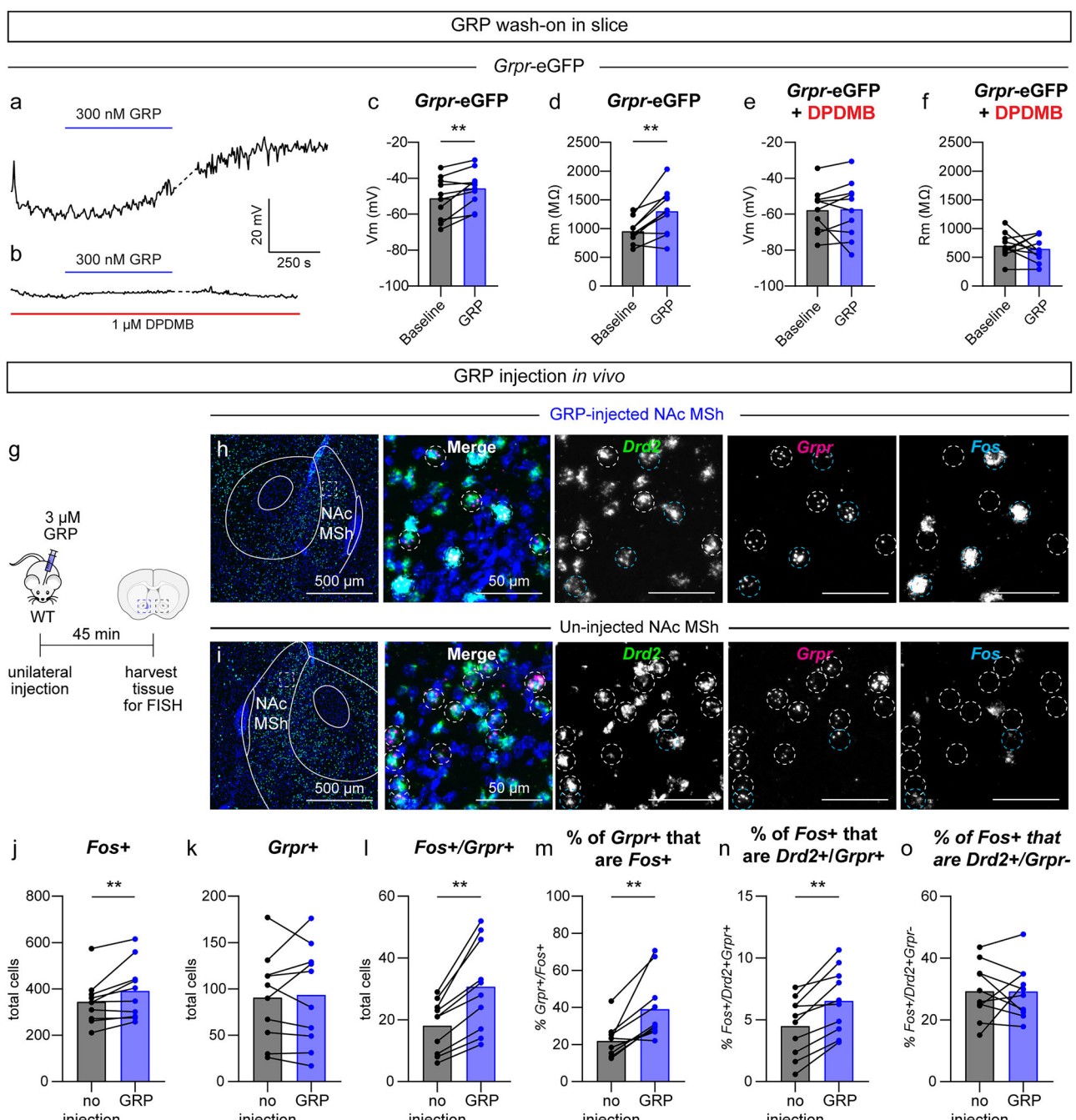

**Fig. 6 | GRP increases the excitability of GRPR-expressing neurons in the NAc MSh. a** Representative trace showing the membrane potential over time of a NAc MSh *Grpr*-eGFP+ cell (representative of 10 neurons). The blue bar represents a 10-minute wash-on of 300 nM GRP. **b** Representative trace from a *Grpr*-eGFP+ cell in which GRP was washed-on in the presence of DPDMB (red bar, entirety of recording, representative of 10 neurons). **c** Membrane potential of *Grpr*-eGFP+ neurons at baseline (black) and after 300 nM GRP wash-on (blue). Baseline value is the average of the 2-minute period prior to GRP wash-on and 300 nM GRP value is the average of the final two minutes of the 10-minute GRP wash-on. **d** Membrane resistance of *Grpr*-eGFP+ neurons at baseline and after GRP wash-on. Baseline value is the average of five sweeps of VC RC check at the beginning of the recording. **e** Membrane potential of *Grpr*-eGFP+ neurons at baseline and after GRP wash-on in the presence of DPDMB. **f** Membrane resistance of *Grpr*-eGFP+ neurons at baseline and after GRP wash-on in the presence of DPDMB. **g** Schematic of the experiment.

**h, i** Representative images of the NAc MSh from the **h** injected hemisphere and **i** un-injected hemisphere. Images are representative of 20 images taken from 10 mice. FISH for *Drd2* in green, *Grpr* in magenta, and *Fos* in cyan. Nuclei are labeled with DAPI in blue. Dashed circles outline *Grpr*+ cells, white=*Grpr*+/*Fos*-, cyan=*Grpr*+/*Fos*+. **j** Total number of cells in the NAc MSh that were *Fos*+ in the un-injected (black) vs GRP-injected hemisphere (blue). **k** Total number of cells that were *Grpr*+. **l** Total number of *Grpr*+ cells that were *Fos*+. **m** Percentage of *Grpr*+ cells that were *Fos*+. **n** Percentage of *Fos*+ cells that were *Drd2*+ and *Grpr*+. **o** Percentage of *Fos*+ cells that were *Drd2*+ and *Grpr*-. For panels **c**–**f** and **j**–**o**, bars represent the mean and dots represent values for individual mice. Values in panels **j**–**o** were obtained by counting all cells in the NAc MSh from two independent slices per mouse that were summed together, *n* = 10 mice. Source data are provided as a Source Data file. See the Supplementary Data file for sample sizes and statistics.

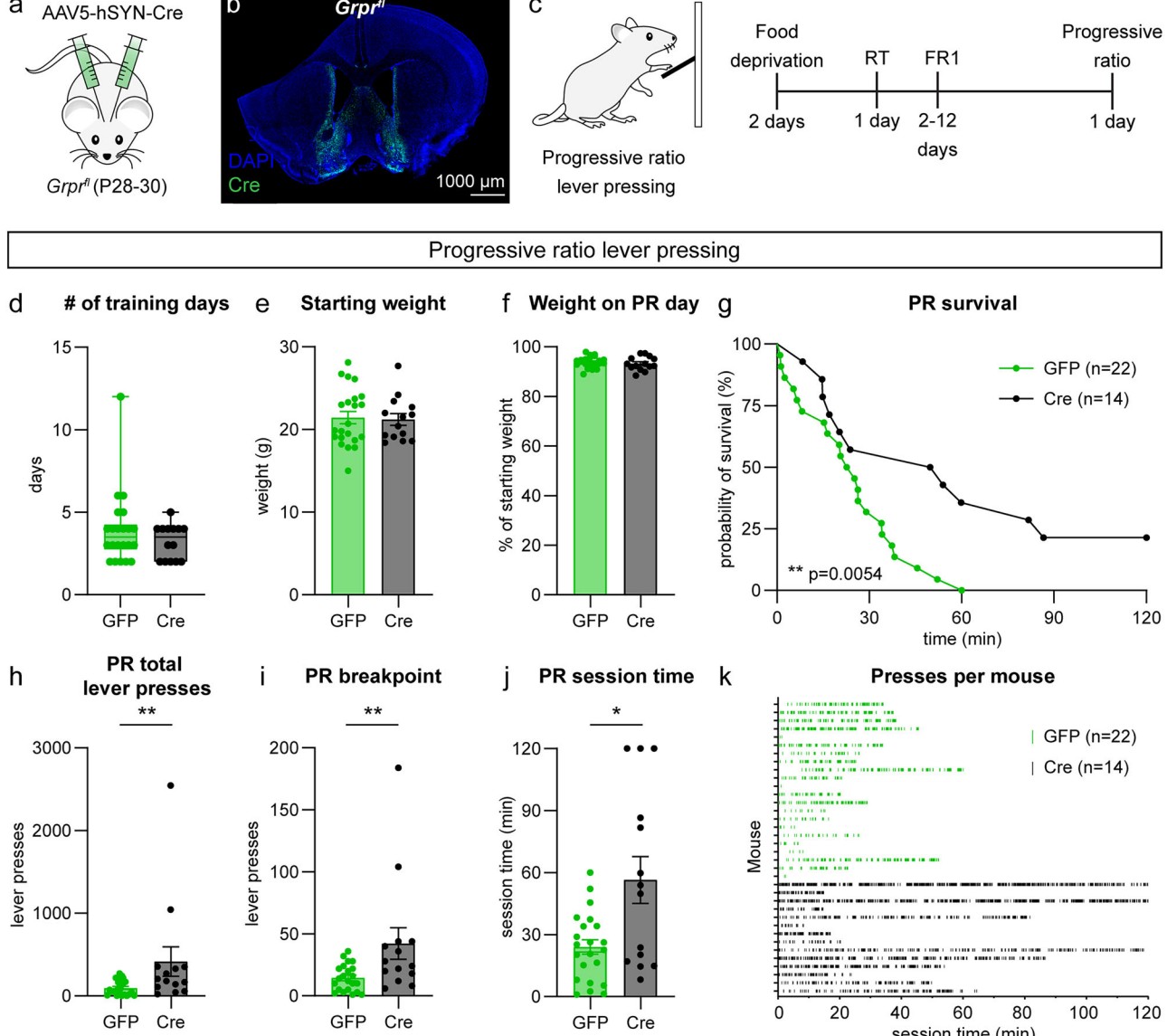

**Fig. 7 | *Grpr* deletion in the NAc MSh enhances motivation in the progressive ratio test. a** Schematic of the experiment. *Grpr*^fl/fl or fl/y^ mice were injected bilaterally with AAV5-hSYN-GFP or AAV5-hSYN-Cre-GFP in the NAc MSh. **b** Sample image of the injection site in the NAc MSh showing AAV-Cre-GFP in green with nuclei labeled in blue with DAPI (representative of 14 mice). **c** Schematic of the progressive ratio (PR) lever pressing test and timeline for operant conditioning. RT = random timing, FR1 = fixed ratio 1. **d** Box plots of the number of days it took control (GFP-injected, green) and *Grpr* cKO (Cre-injected, black) mice to reach criterion (boxes represent the interquartile range (25–75%), lines denote the median, whiskers represent mix to max values) (two-tailed Mann-Whitney test, *p* = 0.4522; *n* = 22 GFP and *n* = 14 Cre mice). **e** Mean ± SEM weight of GFP and Cre-injected mice just prior to starting food deprivation (two-tailed Mann-Whitney test, *p* = 0.8917, *n* is the same as panel (**d**). **f** Mean ± SEM percentage of starting weight per mouse on the day the mouse

completed the PR task (two-tailed Mann-Whitney test, *p* = 0.4192). **g** Survival function of total session time on PR testing day. The session ended when a mouse went three minutes without pressing the lever after pressing the lever at least one time (Mantel-Cox test, **p* = 0.0054). **h** Mean ± SEM total lever presses in the PR session before timeout (two-tailed Mann-Whitney test, **p* = 0.0077). **i** Mean ± SEM breakpoint in the PR session before timeout (two-tailed Mann-Whitney test, **p* = 0.0060). **j** Mean ± SEM total session time during the PR task before timeout (two-tailed Mann-Whitney test, **p* = 0.0381). **k** Raster plot of lever presses per individual mouse during the PR task. Each line represents a single lever press. Each row represents the lever-pressing data of an individual mouse. For all bar graphs, dots represent values for individual mice. For panels d-k, *n* = 22 GFP-injected (green, control) and *n* = 14 Cre-injected (black, *Grpr* cKO) mice. Source data are provided as a Source Data file.

and show that this reduces the baseline activation of the *Grpr*+ neurons and leads to increased motivation in a PR task.

### Grpr defines subtypes of SPNs within the NAc MSh
The NAc MSh has canonically been thought to comprise two main types of SPNs that together make up 90-95% of the cells in the region[19]. Traditionally, SPNs were divided into indirect- and direct-pathway neurons based on the type of DA receptor they expressed and their output to other basal ganglia structures[2,18,68–70]. dSPNs express D1

receptors, dynorphin, and substance P and have been thought to promote motivated behaviors[27,71,72]. In contrast, iSPNs express D2 receptors, A2A receptors, and enkephalin and generally inhibit motivated behaviors[27,71,72]. However, a scRNA-seq study identified a third type of SPN, eSPNs[21]. These cells were defined as SPNs based on their expression of classical SPN markers such as *Ppp1r1b*, *Adora2a*, and *Drd1*, but were found to have a unique genetic profile[21]. In addition, a recent study examining a different subpopulation of neurons in the NAc MSh that express *Crh*, also found these cells to be enriched in

eSPN markers. eSPNs have also been identified in the human striatum[73], but have yet to be functionally characterized or compared to their classical SPN counterparts[45].

Here we report that *Grpr*-expressing cells in the NAc MSh represent at least two genetically unique subpopulations, one that co-expresses many canonical iSPN markers including *Drd2, Penk, Gad2*, and *Ppp1r1b* and another that expresses eSPN markers including *Casz1, Th, Otof, Cacng5*, and *Pcdh8*. In terms of morphology, all *Grpr+* cells had spines, and their dendritic morphology was not different from neighboring cells in the NAc MSh, the vast majority of which are expected to be SPNs. However, the electrophysiological properties of *Grpr*-eGFP+ cells were distinct from all other examined cell types in the NAc MSh. While iSPNs in the NAc MSh are the most depolarized and excitable SPNs at baseline[2,74–77], *Grpr*-eGFP+ cells were both more depolarized and had an approximately 160% greater membrane resistance than *Grpr*-negative *Drd2* + SPNs.

Cluster analysis based on the baseline electrophysiological properties of *Grpr+* cells revealed three subpopulations: one that was similar to traditional iSPNs, one that had the profile of immature iSPNs, and a third group that was distinct from both of these, which may represent eSPNs. We found parallel results in vivo as *Grpr+* cells were over twice as likely to be *Fos+* than *Grpr-* iSPNs in the NAc MSh. Interestingly, when we disrupted *Grpr* expression in these cells, *Fos* was no longer preferentially expressed in *Grpr+* compared to *Grpr-* iSPNs, indicating that GRP-GRPR signaling contributes to the baseline excitability of these cells. *Casz1 + /Grpr+* cells were also more likely to be *Fos + * compared to *Casz1 + /Grpr-* cells. However, *Fos* expression persisted in the absence of *Grpr* suggesting that *Casz1 + /Grpr+* cells are more likely to be activated for reasons independent of the expression of *Grpr*. Together, our detailed analysis highlights the molecular and functional heterogeneity of cells within the NAc MSh, warranting further investigation into how these different cell types contribute to the computations and output of the NAc. Notably, while the *Grpr+* cells in the NAc were generally *Drd2 + *, we also observed a population of *Grpr+* cells in the DS that were *Drd1 + *. It would be interesting in future studies to define the properties of these cells and investigate their behavioral relevance.

### Both glutamatergic and dopaminergic inputs into the NAc MSh have Grp

This study was based on prior work showing that subpopulations of midbrain DA neurons express *Grp*[16,78,79] and project to the NAc MSh[5]. Here, we found that additional *Grp*-expressing inputs come into the NAc MSh from the ventral hippocampus and amygdala and are glutamatergic. This raises the possibility of co-release of GRP with either DA or glutamate from multiple distinct inputs. While future investigation will be needed to understand the specific behavioral roles of GRP signaling from different inputs into the NAc, a common theme is that the *Grp*-expressing inputs all positively regulate motivated behavior. For example, activation of the population of medial VTA neurons projecting to the NAc MSh that express *Neurod6* (which also typically express *Grp*[5]) induces place preference[80]. Ventral hippocampal and subiculum inputs to the NAc MSh promote responses to drugs of abuse as well as drug seeking behavior[81,82]. BLA input to the NAc is rewarding, and optogenetic stimulation can drive conditioned place preference[83]. GRP release from these inputs could serve as a built-in break or constraint on motivated behavior. If this is the case, GRP-GRPR signaling may represent a tunable mechanism to control aberrant motivation and reward responses in disease-related contexts.

GRPR is a $G_q$-coupled receptor, and activation leads to an increase in intracellular calcium[84]. Previous studies examining GRP-GRPR signaling in other brain regions found that GRP excites GRPR-expressing neurons in the auditory cortex, paraventricular thalamus, lateral amygdala, hippocampus and spinal cord[9,52–54,85–87]. In the spinal cord, GRP increases neuronal excitability via blockade of $K_{ir}2$ potassium

channels[9]. We found that while the *Grpr+* neurons in the NAc MSh were already spontaneously active at baseline, application of GRP in slice caused further depolarization and increased the membrane resistance by 140%. Importantly, while there are multiple subpopulations of *Grpr+* cells in the NAc, all cells responded to GRP wash-on. These effects could be blocked by a GRPR-specific antagonist and the presence of the antagonist alone led to a decrease in baseline membrane resistance, raising the possibility that there is basal GRP-GRPR activity.

SPNs are signal integrators that can respond to a variety of neurotransmitters and neuropeptides[88]. Since the majority of GRPR-expressing SPNs also have D2Rs, it is interesting to consider how these signaling pathways may interact within NAc neurons. D2Rs are $G_{i/o}$-coupled and are expected to inhibit PKA-dependent phosphorylation of targets that control cellular excitability[89]. This would presumably counteract the excitatory effect of GRPR-activation. One possibility is that differences in the timing of these signals could differentially affect cellular excitability. For example, the effects of GRPR activation would increase neuronal excitability via regulation of ion channels within a few minutes, whereas activation of $G_{i/o}$-coupled pathways may have more prolonged effects via changes in phosphorylation. It is also possible that DA and GRP are released at different axonal sites or in response to different patterns of activity. Therefore, their downstream signaling pathways may not be engaged in the same place at the same time. Understanding how and when different pre-synaptic signals are engaged and how they interact in the target cells is an important area of future investigation.

### GRP-GRPR signaling affects NAc-associated behaviors

GRP-GRPR signaling has distinct effects on behavior depending on the brain region examined[13,14]. For example, *Grpr* KO mice showed enhanced long-term fear memories, thought to reflect changes in amygdala function[52]. In the spinal cord, GRP-GRPR signaling is the molecular basis of itch[9–11,90–92]. More recently, *Grpr* ablation in the auditory cortex decreased auditory fear memories[53]. Here, we selectively deleted *Grpr* from NAc MSh neurons to test whether striatal-associated behaviors were impacted. We did not find major differences in general behavior patterns in the open field, motor learning, or sucrose preference in *Grpr* cKO mice, indicating that GRPR signaling is not required for these behaviors. We did, however, find that *Grpr* cKO mice showed increased motivation in the PR task. We do not believe this is due to increased lever pressing in general, as there were no significant differences in lever-press behavior during FR1 training. The NAc MSh is an important regulator of motivation, and, more specifically, iSPNs have been shown to inhibit motivation[28,64]. In a prior study, chemogenetics was used to inhibit NAc iSPN outputs to the VP, which increased motivation. This is consistent with studies showing that iSPNs typically inhibit motivated behaviors[2,22,27]. Here, by eliminating GRP-GRPR signaling specifically in the NAc MSh, we expected to reduce the excitability of a small subpopulation of iSPNs. Consistent with this, we found that the percentage of *Fos*-positive *Grpr*-lineage iSPNs was reduced in *Grpr* KO mice. The inability to activate *Grpr*-expressing SPNs through GRPR signaling led to increased motivation, consistent with studies broadly manipulating iSPNs in this region. This indicates that GRP-GRPR signaling in the NAc MSh normally constrains motivation and suggests that the relatively small population of GRPR+ NAc neurons can exert significant control over motivated behavior.

### Limitations and future directions

This study identified and characterized previously undescribed sub-populations of SPNs in the NAc MSh that express *Grpr*. We provide a genetic, morphological, and electrophysiological characterization of these cells and identify a behavioral role of GRPR signaling in this region. These experiments lay the groundwork for further characterization of GRP-GRPR signaling in the striatum and in the brain, in general. We note that although it was beyond the scope of the present

study, *Grpr+* cells also exist in the DS. Given the unique electrophysiological properties of the GRPR+ cells in the NAc MSh, the GRPR+ cells in the DS warrant their own characterization. Interestingly, there does not appear to be an eSPN population of *Grpr+* cells in the DS in mice[30]. Additionally, while we identified the *Grp+* inputs into the NAc MSh and noted that the *Grpr+* cells themselves were GABAergic, we were unable to address whether the *Grpr+* cells project to and inhibit the VP. This was largely due to limitations in the available transgenic mouse lines for manipulating *Grpr*-expressing cells. Recent evidence has indicated that unlike the DS, the canonical projection pathways from the NAc are oversimplified and both D1- and D2-SPNs can participate in indirect and direct pathways depending on their downstream target from the VP[28]. Therefore, additional tools are needed to examine the outputs of *Grpr+* cells.

While we performed cellular-level characterization of the *Grpr*-expressing cells and tested a subset of behaviors that have been associated with the NAc MSh, future studies could assess a possible role of GRP-GRPR signaling in additional behaviors. One such direction could utilize tasks that assess context-induced reinstatement of drug seeking for drugs of abuse. These tests have been specifically associated with ventral subiculum to NAc MSh projections, where we find a large population of *Grp+* inputs to the NAc MSh[81,93]. Finally, given the $G_q$-coupling of GRPR, a possible role of GRP in modulating synaptic transmission or plasticity could be examined.

Overall, our study identifies previously undescribed cell types in the NAc MSh and highlights a mesolimbic circuit defined by the expression and functionality of GRP to GRPR signaling.

# Methods

## Mice
Animal experiments were performed in accordance with protocols approved by the University of California, Berkeley, Institutional Animal Care and Use Committee (protocol #: AUP-2016-04-8684-2). Both male and female mice were used across all experiments except for the global *Grpr* knock-out FISH experiments, where male littermate WT and *Grpr* KO mice were used. Experiments were not designed to test for sex differences. Mice were P30-P90 unless otherwise stated. The following mouse lines were used in this study: C57BL/6 J (JAX strain #000664), *Grpr*-eGFP (MMRRC #036178-UCD)[33], *Drd2*-eGFP (MMRRC #000230-UNC)[33], *Drd1*-tdTomato (JAX strain #016204)[35], *Adora2a*-Cre (MMRRC #036158-UCD)[36], CMV-Cre (JAX strain #006054)[50], *Grpr*fl (JAX strain #033148)[11], *Chat*-IRES-Cre (JAX strain #06410)[39], and Ai9 (JAX strain #07909)[37]. *Grpr* KO mice were generated by breeding CMV-Cre mice to *Grpr*fl mice to knock out *Grpr* in all cells. The number of mice used in each experiment is indicated in the figure legend or Supplementary Data file.

## Fluorescent in situ hybridization
Fluorescent in situ hybridization was performed to identify and characterize *Grpr*-expressing cells in the NAc MSh and DS as well as their *Grp*-expressing inputs. Mice were anesthetized with isoflurane using the drop jar method. Brains were harvested, flash-frozen in OCT mounting medium (Fisher Scientific #23-730-571) on dry ice and stored at −80 °C for up to 6 months. 16 μm-thick coronal sections were collected using a cryostat, mounted directly onto 75 x 25mm Superfrost® Plus glass slides (VWR #48311-703) and stored at −80 °C for up to 1 month. Fluorescent in situ hybridization was performed according to protocols provided with the RNAscope® Multiplex Fluorescent Reagent Kit (ACD #320850) and RNAscope® Multiplex Fluorescent Reagent Kit V2 (ACD #323100). *Grpr* mRNA was visualized with a probe in channel 2 (ACD #317871-C2), *Gad2* mRNA in channel 1 (ACD #439371), *Drd2* mRNA in channel 1 (ACD #406501), *Drd1* mRNA in channel 3 (ACD #461901-C3), *Chat* mRNA in channel 1 (ACD #408731), *Ppp1r1b* mRNA in channel 3 (ACD #405901-C3), *Penk* in channel 3 (ACD # 318761-C3), *Casz1* mRNA in channel 3 (ACD #502461-C3), *Fos* mRNA in

channel 1 or 3 (ACD #316921 or ACD #316921-C3), *Grp* mRNA in channel 2 (ACD #317861-C2), and *Slc17a7* mRNA in channel 1 (ACD #481851).

Sections were imaged using an Olympus FluoView 3000 confocal microscope equipped with 405, 488, 561, and 640 nm lasers and a motorized stage for tile imaging. Z stack images captured the entire thickness of the section at 1-2.5 μm steps for images taken with 20X (Olympus #UCPLFLN20X) or 10X air objectives (UPLXAPO10X). Additional images were acquired on a Zeiss LSM 710 AxioObserver with Zeiss 10X, 20X, and 63X objectives housed in the Molecular Imaging Center at UC Berkeley. Images were analyzed using FIJI. Cells were considered positive for *Grpr* if *Grpr* mRNA covered at least one percent of the nucleus as defined by DAPI. All images for a single dataset were acquired on the same microscope. Quantifications represent the summed values from all cells within the entire NAc MSh of two independent sections per mouse.

## Immunohistochemistry
Male and female mice were deeply anesthetized by isoflurane and transcardially perfused with ice cold 1X PBS (∼5–15 mL) followed by 4% paraformaldehyde (PFA) solution (Electron Microscopy Sciences: 15713) in 1X PBS (∼5–15 mL) using a peristaltic pump (Instech). The brains were removed and post-fixed by immersion in 4% PFA in 1X PBS solution overnight at 4 °C. Brains were suspended in 30% sucrose in PBS (1X pH 7.4) for cryoprotection. After brains descended to the bottom of the vial (typically 24–48 h), 40 μm coronal sections of the NAc MSh were cut on a freezing microtome (American Optical AO 860), collected into serial wells, and stored at 4 °C in 1X PBS containing 0.02% (w/v) sodium azide (NaN3; Sigma Aldrich).

Free-floating sections were washed with gentle shaking, 3 × 5 min in 1X PBS followed by 1 hr incubation at RT with BlockAid blocking solution (Life Tech: B10710). Primary antibodies were applied at 4 °C in 1X PBS containing 0.25% (v/v) triton-X-100 (PBS-Tx) overnight to ensure penetration of the antibody throughout the slice. Sections were then washed with cold 1X PBS-Tx 3 × 10 min and incubated for 2 hr at RT with secondary antibodies in PBS-Tx. Sections were washed in cold 1X PBS 3 × 10 min, mounted on SuperFrost slides (VWR: 48311–703), and coverslipped with Prolong Gold antifade mounting media with DAPI (Life Tech: P36935).

We used a primary antibody against GFP (1:5000, Chicken, Abcam: 13970) and an Alexa Fluor 488 goat anti-chicken secondary antibody (1:500, Invitrogen: A-11039).

Images of 40 μm sections processed for immunohistochemistry were acquired using an Olympus FluoView 3000 confocal microscope (described above). Z-stack images captured the entire thickness of the section at 1–2 μm steps for images taken with a 20X air (Olympus #UCPLFLN20X) or a 10X air Olympus (#UPLXAPO10X) objective.

## Electrophysiology
Mice (P8-12; P30-90) were anesthetized with isoflurane using the drop jar method and then briefly perfused transcardially with ice-cold ACSF (pH = 7.4) containing (in mM): 127 NaCl, 25 NaHCO3, 1.25 NaH2PO4, 2.5 KCl, 1 MgCl2, 2 CaCl2, and 25 glucose, bubbled continuously with carbogen (95% O2 and 5% CO2). Brains were rapidly removed, and coronal slices (275 μm) were cut on a VT1000S vibratome (Leica) in oxygenated ice-cold choline-based external solution (pH = 7.8) containing (in mM): 110 choline chloride, 25 NaHCO3, 1.25 NaHPO4, 2.5 KCl, 7 MgCl2, 0.5 CaCl2, 25 glucose, 11.6 sodium ascorbate, and 3.1 sodium pyruvate. Slices were recovered in ACSF at 34 °C for 15 min and then kept at RT before recording.

Recordings were made with a MultiClamp 700B amplifier (Molecular Devices) at RT using 3-5 MΩ glass patch electrodes (Sutter: BF150-86-7.5). Data were acquired using ScanImage software, written and maintained by Dr. Bernardo Sabatini (https://github.com/bernardosabatini/%20SabalabAcq). Traces were analyzed in Igor Pro

(Wavemetrics). Recordings with a series resistance > 30 MΩ were rejected.

**Current-clamp recordings.** Current clamp recordings were made using a potassium-based internal solution (pH=7.4) containing (in mM): 135 KMeSO$_4$, 5 KCl, 5 HEPES, 4 Mg-ATP, 0.3 Na-GTP, 10 phosphocreatine, and 1 EGTA. No synaptic blockers were included, and no holding current was applied to the membrane. For intrinsic excitability experiments, positive current steps (1 s, +10 to +100 pA) were applied in 10 pA intervals to generate an input-output curve. Resting membrane potential (Vm) was calculated from current clamp traces recorded for the input-output experiment. Vm was calculated as the average membrane potential over a 300-ms baseline period between 50–350ms of a 2-second recording. The plotted value for resting membrane potential is the average of this value for all traces over a two-minute (min) period. Membrane resistance was calculated from voltage-clamp RC checks prior to switching to current-clamp. A cell was considered spontaneously active at baseline if it fired at least one action potential during the first two min after breaking in without current injection. Action potential half width was calculated as the width at half maximum amplitude of spikes during the first current step that induced spiking. All action potential properties were from the first action potential a cell fired after current injection.

**GRP wash-on.** Membrane potential changes upon bath application of 300 nM GRP (Anaspec: AS-24214) dissolved in water and diluted in ACSF were monitored in current-clamp mode. 5 s current-clamp recordings were made every 2 s. Prior to GRP wash-on, membrane potential was recorded for 10 min at baseline, and membrane potential was averaged over a 300 ms window between 50–350 ms of the 2 s sweep. 300 nM GRP was then bath applied for 10 min. At the conclusion of GRP wash-on, we switched back to voltage-clamp to perform 5 RC checks to calculate membrane resistance and confirm that series resistance was stable and <30 MΩ. GRP was then washed out. The baseline membrane potential was calculated as the average over the two minutes prior to GRP wash-on. The membrane potential following GRP application was calculated as the average over the last two minutes of wash-on. Baseline membrane resistance was calculated from the initial voltage-clamp RC check. Wash-on experiments in the presence of the GRPR antagonist, 1 μM (D-Phe[6], Leu-NHEt[13], des-Met[14])-Bombesin (6-14) (DPDMB, Bachem: H-3042) were performed in the same way, except the slices were pre-incubated in DPDMB for at least 15 min prior to recording and DPDMB was present in the ACSF for the duration of the recordings.

**Neurobiotin-filled neuron reconstruction for Sholl analysis**
Male and female mice were deeply anesthetized by isoflurane, transcardially perfused with ice-cold ACSF using a peristaltic pump (Instech), and decapitated. 275-μm-thick coronal striatal slices were prepared on a vibratome (Leica VT1000S) in oxygenated ice-cold choline-based external solution (pH 7.8) containing 110 mM choline chloride, 25 mM NaHCO$_3$, 1.25 mM NaHPO$_4$, 2.5 mM KCl, 7 mM MgCl$_2$, 0.5 mM CaCl$_2$, 25 mM glucose, 11.6 mM sodium ascorbate, and 3.1 mM sodium pyruvate. Slices were recovered in ACSF at 34 °C for 15 min and then kept at room temperature (RT) for the duration of the recordings. All solutions were continuously bubbled with 95% O$_2$ and 5% CO$_2$. For whole cell recordings, 3–5 mΩ borosilicate glass pipettes (Sutter Instrument: BF150-86-7.5) were filled with a potassium-based internal solution (pH 7.4) containing 135 mM KMeSO$_4$, 5 mM KCl, 5 mM HEPES, 4 mM Mg-ATP, 0.3 mM Na-GTP, 10 mM phosphocreatine, 1 mM EGTA, and 4 mg/mL neurobiotin (Vector Laboratories: SP-1120).

*Grpr*-eGFP positive or negative NAc MSh neurons were filled with neurobiotin-containing internal solution (4 mg/mL) during patch-clamp experiments and then fixed in 4% paraformaldehyde solution (Electron Microscopy Sciences: 15713) in 1X PBS for 24–48 h at 4 °C.

With continuous gentle rocking, slices were washed in 1X PBS 3 × 5 min and incubated with BlockAid blocking solution (Life Tech: B10710) for 1.5 hr at RT. Streptavidin Alexa Fluor 633 conjugate (1:2000, Invitrogen: S21375) and Streptavidin Alexa Fluor 488 (1:1000, Invitrogen: S32354) were applied for 2 h in 1X PBS containing 0.50% (v/v) triton-X-100 (PBS-Tx). Slices were then washed in 1X PBS 3 × 10 min, mounted on SuperFrost slides (VWR: 48311–703) with the cell-containing side of the slice facing up, and coverslipped with Prolong Gold antifade mounting media with DAPI (Life Tech: P36935).

Filled cells in the mounted sections were imaged on an Olympus FluoView 3000 confocal microscope at 20X magnification with 2X digital zoom for a final magnification of 40X. 1 μm steps were used to acquire a z-stack spanning the entirety of the neurobiotin-filled cell body and dendritic arbors. 3D reconstructions of the cells were done using IMARIS 9.2.1 software (Bitplane) with automated filament tracing and manual editing. The mask generated by the automated filament tracing algorithm was continuously cross-referenced with the original z-stack image to ensure accuracy. Spurious segments created by the automated filament tracer were removed, while processes with incomplete reconstruction were manually edited to incorporate missing segments.

**Sparse viral injections for spine density quantification**
Neonatal (P0) or juvenile (P28-30) *Grpr*-eGFP mice were injected bilaterally with 250 nL (neonates) or 100 nL (juveniles) AAV5-hSyn-mCherry (Addgene: 114422-AAV5) diluted 1:200 to sparsely label neurons in the NAc MSh. Neonatal mice were cryoanesthetized and injected bilaterally at 200 nL/min. Injections were targeted to the NAc MSh, with coordinates approximately 0.75 mm lateral to midline, 1.0 mm posterior to bregma, and 3.0 mm ventral to the head surface. For juvenile injections, mice were anesthetized with 3% isoflurane (Piramal Healthcare: PIR001710) and oxygen. Mice were then mounted on a stereotaxic frame (Kopf instruments Model 940) with stabilizing ear cups and a nose cone delivering constant 1.5% isoflurane in medical oxygen. Viruses were injected using a pulled glass capillary (Drummond Scientific Company: 5-000-2005) at a rate of 100 nL/min. Following injection, the capillary remained in place for 1 min per every min spent injecting to allow the tissue to recover and prevent virus backflow up the injection tract upon retraction. Coordinates from bregma for juvenile injections were as follows: M/L + /−1.00 mm, A/P + 1.90 mm, D/V −4.50 mm.

At P56, mice were anesthetized with isoflurane using the drop jar method and perfused transcardially with 1x PBS followed by 4% paraformaldehyde. Brains were post-fixed with 4% paraformaldehyde overnight, then sectioned at 80 μm. Sections were blocked for 1 h at RT in BlockAid (ThermoFisher: B10710) and incubated for 16 h at 4 °C with antibodies against RFP (1:500, rabbit, Rockland (VWR): RL600-401-379) and GFP (1:500, chicken, Abcam: 13970). Sections were washed 3 × 10 min in PBS-Tx and incubated for 2 h at RT with 1:500 Alexa Fluor 488 goat anti-chicken (ThermoFisher: A-11039) and 1:500 Alexa Fluor 546 goat anti-rabbit secondary antibodies (ThermoFisher: A-11035). Sections were washed 3 × 10 min in 1X PBS and mounted onto slides using Prolong Gold antifade mounting media with DAPI (Life Tech: P36935).

Dendrites were imaged on an Olympus FluoView 3000 scanning confocal microscope at 60X magnification with 2.5X digital zoom for a final magnification of 150X. Images were deconvoluted in the CellSens4 software using a built-in advanced maximum likelihood algorithm. Dendritic spine reconstructions were generated using the filament tracer feature in IMARIS 9.3.1. 1-9 dendrites were reconstructed per cell, with the detection parameters for thinnest spine diameter set to 1.5 μm and the maximum spine length set to 3.5 μm. After automatic detection of spines, the digital reconstruction was compared to the original z-stack image to ensure accuracy. Detected spines were manually reviewed by the experimenter and added or

removed as needed. Spine density of each dendrite was calculated by dividing the number of spines on that dendrite by the total dendrite length. The values of all dendrites quantified for a given neuron were averaged and the average spine density per dendrite is reported.

## Single-cell RNAseq analysis

scRNAseq data was downloaded from the Allen Brain Cell Atlas (Mouse[30]:, Human[45]:). Expression matrices and metadata for relevant brain regions were extracted, processed, and analyzed using Seurat v.5.1.0[94]. For mouse, data corresponding to the ventral (STRv) and dorsal (STRd) striatum were selected. For human, data corresponding to the dissection region "Basal nuclei (BJ) - Nucleus Accumbens - NAC" were selected. Striatum datasets were processed using the "Standard Seurat Workflow" (https://satijalab.org/seurat/articles/essential_commands.html). Subsetted Seurat objects were re-processed using this workflow before additional analysis and plotting. To produce GRPR-only datasets, any cell expressing non-zero levels of *Grpr/GRPR* were subsetted from the overall Seurat objects and re-processed using the Standard Seurat Workflow. Data for gene expression violin plots were extracted from the log-normalized RNA assay. Code for the scRNAseq analysis is available at https://github.com/BateupLab/Aisenberg2025[95].

## Passive properties clustering analysis

Passive properties were visualized using the pairplot function in the Seaborn Python package[96]. Passive properties were analyzed using UMAP-learn[97] and Scikit-learn[98] Python packages. Briefly, data was pre-processed using the StandardScaler function, and UMAP embeddings were calculated using the following parameters: (n_neighbors=10, min_dist=0, metric = 'euclidean'). Clustering was performed on the scaled dataset using the K-means approach with an n_clusters value of 4.

## Intracranial injections

**Retrobead injections.** P50-90 WT male and female mice were used for retrograde labeling experiments. Mice were anesthetized with 3% isoflurane and oxygen and mounted on a Kopf stereotaxic frame with stabilizing ear cups and a nose cone delivering constant 1.5% isoflurane in medical oxygen. Red or green retrobeads IX (Lumafluor) were diluted 1:3 in sterile saline. 350 nL diluted beads were unilaterally injected into the NAc MSh using a pulled glass pipette. The following coordinates from bregma were used: M/L + 0.65 mm, A/P + 1.90 mm, D/V −4.50 mm. To allow for sufficient labeling, mice were sacrificed 14 days post injection. Mice were anesthetized with isoflurane using the drop jar method. Brains were then harvested, flash-frozen in OCT mounting medium (Fisher Scientific: 23-730-571) on dry ice, and stored at −80 °C for up to 6 months. 16 μm-thick coronal sections were collected using a cryostat, mounted directly onto 75 x 25mm Superfrost® Plus glass slides (VWR: 48311-703) and stored at −80 °C for up to 1 month. Prior to sectioning the posterior part of the brain, the injection site was validated to ensure that retrobeads were constrained to the NAc MSh. After confirming, the rest of the brain was sectioned and fluorescent in situ hybridization was performed on all sections containing brain regions with retrobead+ cells.

**GRP injections.** Male and female mice underwent stereotaxic surgery to inject GRP into the NAc MSh. Mice were anesthetized with 3% isoflurane and oxygen and mounted on a Kopf stereotaxic frame with stabilizing ear cups and a nose cone delivering constant 1.5% isoflurane in medical oxygen. 800 nL of 3 μM GRP (dissolved in sterile saline) or sterile saline were injected unilaterally into the left NAc MSh. The stereotaxic coordinates used were: +0.65 mm, A/P + 1.90 mm, D/V −4.50 mm. 45 min after the end of the injection, mice were anesthetized with isoflurane, and their brains harvested and flash frozen for fluorescent in situ hybridization.

**AAV-Cre and AAV-GFP injections.** Male and female mice (P28-30) underwent stereotaxic surgery to inject pENN.AAV.hSyn.HI.eGFP-Cre.WPRE.SV40 (AAV5) or pAAV-hSyn-EGFP (AAV5) into the NAc MSh. 600 nL of 1:3 diluted Cre or GFP virus in sterile saline were injected bilaterally into the NAc MSh. The stereotaxic coordinates used were: +/−0.85 mm, A/P + 1.70 mm, D/V −4.50 mm. Mice were allowed to recover and four weeks later, mice underwent behavioral testing. At the end of behavioral testing, mice were perfused and the tissue was sliced to confirm proper viral targeting.

## Genotyping of global KO mice

Multiple PCRs were used as a genotyping strategy to confirm the genotype of *Grpr* KO and WT control mice. The strategy involved primer pairs for the presence of the floxed or WT allele, the LoxP scar, and an internal region of *Grpr* exon 2. The following primer sets were used: *Grpr* floxed (WT-forward: 5′ GGAGAGAGGTATAGAGGGGC 3′, WT-reverse: 5′ ACATGTAGGTTGCAGGGGAT 3′, MUT-forward 5′ GGGTTATTGTCTCATGAGCGG 3′), LoxP scar (Forward 5′ AGACAGCT CTATCACGGTCC 3′, Reverse 5′ GAGTTCAGTTCCCAGCAACC 3′), *Grpr* Exon 2 (Forward 5′ TTTCTGACCTCCACCCCTTC 3′, Reverse 5′ CACGG GAAGATTGTAGGCAC 3′). With this strategy, we expected the following bands for *Grpr* KO mice: *Grpr* floxed (no band), LoxP scar (298 bp), *Grpr* Exon 2 (no band) and WT mice: *Grpr* floxed (573 bp WT band), LoxP scar (no band), *Grpr* Exon 2 (206 bp).

## Behavioral experiments

Behavior studies were carried out in the dark phase of the light cycle under red lights (open field) or white lights (rotarod and operant conditioning). Sucrose preference was carried out in the room where the mice are normally housed. Mice were habituated to the behavior testing room for at least 30 min prior to testing, and at least 24 hours elapsed between sessions. All behavioral equipment was cleaned between each trial with 70% ethanol. Equipment was rinsed with diluted soap and water at the end of each day. Behavioral experiments included both male and female mice and began four weeks after the stereotaxic injections at P54-P60. Male mice were trained or tested before female mice each day. Experimenters were blinded to the genotype during behavioral testing.

## Open field

**Open field.** Exploratory behavior in a novel environment and general locomotor activity were assessed by a 60 min session in an open field chamber (40 cm L x 40 cm W x 34 cm H) made of transparent plexiglass. The mouse was placed in the bottom right hand corner of the arena and behavior was recorded using an overhead monochrome camera (FLIR Grasshopper 3: GS3-U3-41C6NIR-C) with a 16 mm wide angle lens (Kowa: LM16HC) placed on top of the arena from a height of 50 cm. Data were analyzed using DeepLabCut (DLC)[55,99] and Keypoint-MoSeq[56,57].

**Processing and analysis of open field data.** To extract the body part (keypoint) coordinates from the video recordings, we used DLC 2.3.4[55,99]. Fourteen body parts, including nose, head, left ear, right ear, left forelimb, right forelimb, spine 1, spine 2, spine 3, left hindlimb, right hindlimb, tail 1, tail 2, and tail 3, were manually labeled on a small subset of the video frames. A DLC model was then trained using the annotated frames to label these 14 body parts for all videos recorded. The distance traveled and center entries were calculated with the coordinates of the body part tail 1.

Discrete behavior syllables were extracted with Keypoint-MoSeq 0.4.4[57]. Syllable usage and transition data were obtained using built-in functions in the Keypoint-MoSeq package. Decoding analysis was done with customized Python 3.9 script. Code for the open field analysis is available at https://github.com/BateupLab/Aisenberg2025[95].

**Rotarod.** The accelerating rotarod test was used to examine motor learning. Mice were tested on a rotarod apparatus (Ugo Basile: 47650) for four consecutive days. Three trials were completed per day with a 5 min break between trials. The rotarod was accelerated from 5–40 revolutions per min (rpm) over 300 s for trials 1–6 (days 1 and 2), and from 10–80 rpm over 300 s for trials 7–12 (days 3 and 4). On the first testing day, mice were acclimated to the apparatus by being placed on the rotarod rotating at a constant 5 rpm for 60 s and returned to their home cage for 5 min prior to starting trial 1. Latency to fall, or to rotate off the top of the rotarod barrel, was measured by the rotarod stop-trigger timer.

**Operant conditioning.** Motivated behavior was assessed in an operant conditioning lever pressing task using a food pellet reinforcer. Animals were food restricted to ~90-95% of their body weight before commencing training in an operant conditioning chamber (Med Associates: ENV-307A). The chambers contained a single retractable lever on the left side of the food receptacle and a house light on the opposite end of the chamber. Each session began with illumination of the house light and the presentation of the left lever. The session ended with retraction of the lever and with the house light turning off. The lever remained present throughout the session. Mice were weighed before each session and habituated to the experiment room for at least 30 min prior to testing.

During the first training session, following two days of food restriction, a food pellet (14 mg regular chow pellet, Bio-Serve: F05684) was delivered on a random time schedule (RT), with a reinforcer delivered on average every 60 s for a total of 15 min with no levers present. On subsequent days, mice underwent 60-min fixed ratio (FR1) sessions where mice were presented with only the left lever. Each lever press was rewarded with a single pellet reward. During initial FR1 sessions, crushed pellets were placed on the lever at the beginning of the task and each time 10 min elapsed without a lever press to prime the mice to press the lever. Mice underwent daily FR1 sessions until they had two consecutive sessions with 20 or more lever presses. After earning 20 rewards the first time, the levers were no longer primed with crushed pellets. The mice then moved on to the progressive ratio (PR) session the following day. The PR session ended when a mouse failed to press the lever for a period of at least three min, after pressing the lever at least once, or after two hours had elapsed, whichever came first. The number of lever presses required for a reward increased with each reward. The number of lever presses to earn a reward were as follows: 1, 2, 3, 4, 5, 6, 7, 8, 10, 12, 14, 16, 18, 20, 22, 24, 28, 32, 36, 40, 44, 48, 52, 56, 64, 72, 80, 88, 96, 104, 112, 120, 128, 136.

**Sucrose preference test.** *Grpr* cKO and GFP-injected control mice were assessed for normal hedonic behavior by using the sucrose preference test. Three days prior to the sucrose preference test, the water source in the home cage was switched from a Hydro Pac (plain water) to a water bottle containing the contents of the Hydro Pac. After three days of acclimating to the water bottles, a second bottle was added to the home cage containing 5% sucrose in Hydro Pac water. Mice had access to both bottles for 24 hours. After 24 hours, mice were individually housed, again with two bottles, one with 5% sucrose and the other with standard water. Mice were kept singly housed for another 24 hours. The volume of water and sucrose consumed during this 24-hour period was measured. Sucrose preference was calculated by taking the total volume of sucrose solution consumed and dividing it by the total liquid consumed in the 24-hour period. To avoid any confounds due to side-specific preferences, the bottles' location was randomly assigned (front or back) for each mouse. All experiments were conducted on weekends, ensuring a quiet environment for the tested animals.

**Quantification and statistical analysis**
GraphPad Prism 10 was used to perform statistical analyses. All datasets were first analyzed using D'Agostino and Pearson normality test, and then parametric or non-parametric two-tailed statistical tests were employed accordingly to determine significance. If the variances between two groups were significantly different, a Welch's correction was applied. Significance was set as $*p < 0.05$, $**p < 0.01$, $***p < 0.001$, and $****p < 0.0001$. All $p$ values were corrected for multiple comparisons. For all experiments, individual values represent biological replicates. Statistical details, including sample sizes for each experiment, are reported in the figure legends or the Supplementary Data File.

**Reporting summary**
Further information on research design is available in the Nature Portfolio Reporting Summary linked to this article.

## Data availability
Source data are provided with this paper in the Source Data file. The mouse striatum single-cell RNA sequencing data are available in from the Allen Brain Cell Atlas under the accession code "WMB-10Xv3" (https://alleninstitute.github.io/abc_atlas_access/descriptions/WMB-10Xv3.html). The human nucleus accumbens single-cell sequencing data are available from the Human Brain Cell Atlas v1.0 at the CELLxGENE database under the name "Dissection: Basal nuclei (BN) - Nucleus Accumbens - NAC" (https://cellxgene.cziscience.com/collections/283d65eb-dd53-496d-adb7-7570c7caa443). Source data are provided with this paper.

## Code availability
Code for the analysis of open field behavior data and single-cell RNA sequencing data is available at: https://github.com/BateupLab/Aisenberg2025[95]. DOI: 10.5281/zenodo.15725043.

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

## Acknowledgements

This work was supported by a Chan Zuckerberg Biohub Investigator grant to H.S.B. H.S.B. is a Weill Neurohub Investigator. We thank Dr. Daniel Kramer for his contribution to the early phases of the project. We thank Dr. Hendrik Wildner and Dr. Hanns Ulrich Zeilhofer for sending us tissue samples for mouse validation. We thank Kamran Ahmed for help with schematics for figures. We thank Dr. Dirk Hockemeyer for help designing a genotyping strategy for the *Grpr* KO mice. Select confocal

imaging experiments were conducted at the CRL Molecular Imaging Center, RRiD:SCR_017852, supported by the Gordon and Betty Moore Foundation. We thank Holly Aaron and Feather Ives for their microscopy advice and support. We thank Mahmoud Farhan for help with mouse colony maintenance and genotyping.

## Author contributions

Conceptualization – E.E.A. and H.S.B.; Formal analysis – E.E.A., T.L.L. and H.W.; Funding acquisition – H.S.B.; Investigation – E.E.A., T.L.L, H.W., A.A.S. and E.M.T.; Methodology – E.E.A., T.L.L., H.W. and E.M.T.; Project administration – E.E.A. and H.S.B., Software – T.L.L. and H.W.; Supervision – H.S.B.; Visualization – E.E.A.; Writing – original draft – E.E.A.; Writing – review & editing – E.E.A., A.A.S. and H.S.B.

## Competing interests

The authors declare no competing interests.
