## [Transparent Peer Review file · Nature Communications]

Gastrin-releasing peptide signaling in the nucleus accumbens medial shell regulates neuronal excitability and motivation

Corresponding Author: Dr Helen Bateup

Version 0:

Reviewer comments:

Reviewer #1

(Remarks to the Author)

Aisenberg and colleagues identified previously undescribed subtypes of spiny projection neurons (SPNs) in the medial shell of the nucleus accumbens (NAc MSh) by using single-cell RNA sequencing and fluorescent in situ hybridization (FISH). They also characterized the electrophysiological properties of Grpr+ neurons. Increased Fos+ labeling supports the electrophysiological findings. Their role in controlling motivated behavior was demonstrated. These findings are a valuable contribution to the understanding of the complex architecture of the ventral stratum. The amount of work is substantial, and the findings provide detailed information on diverse cell types within the NAc and establish molecular markers for Grpr neurons. However, to strengthen the findings, more rigorous electrophysiological recordings and additional approaches to confirm in vivo function will be needed as described below.

Is it possible to examine the Grpr::egfp with D2 mice line to confirm that these are indeed D2-expressing neurons?

Does membrane potential in Figure 1 and Expanded data 2 mean resting membrane potential? How was resting membrane potential measured? Description in the method section is too simple.

In page 7, "Given that Grpr-eGFP+ cells were spontaneously active..", If hyperpolarizing current is injected to lower the membrane potential of Grpr-eGFP+ neurons, do these neurons continue to fire spontaneously?

The sample traces in Extended Data Fig. 3a are identical to those in Fig. 1. Please provide a different example.

To better assess recording quality, please show how the traces look like at each step of current injection throughout the 0-100 mV range (Fig. 1l).

In Figure 4, the representative image shows only a few Fos+ neurons. Counting such a small number in each sample window may lead to biased results. To obtain a more accurate representation of the population, please show and analyze bigger representative images from each sample (i.e., 250 x 250 μ m), providing a clearer view of whether the majority of Grpr+ neurons are Fos+ neurons at the population level.

What percentage of Fos+ neurons are Grpr+ neurons?

In Figure 4, please show the number of neurons analyzed per mouse as well as the total number of neurons.

Was there a decrease in Fos+ neurons in Grpr KO mice?

In Figure 6, test whether GRP application still increases membrane potential at -80 mV when hyperpolarizing current is injected. If it does, by how much does the membrane potential increase?

Please include control experiments; GRP application in Grpr-lineage cells in Grpr KO mice.

Inject GRP into Grpr KO mice to determine whether Fos+ neurons are not increasing.

Discuss potential scenarios in which the membrane potential remains unchanged while the membrane resistance decreases.

In page 8, figure number is missing, (Extended Data Fig. c-f).

Reviewer #2

(Remarks to the Author)

This manuscript primarily examines the characteristics of medium spiny neurons in the nucleus accumbens medial shell that express the gastrin-releasing peptide receptor. Using a combination of RNASeq and FISH, the authors found that there are distinct populations of Penk-lacking D2-MSNs in the NAc mSh that express GRPR. They extensively characterize these neurons in terms of their molecular identity and electrophysiological properties. A major conclusion is that GRPR+ D2-MSNs to comprise a subset of eccentric MSNs. The cellular characterization is excellent (6 of 8 figures). The authors efforts to establish physiological relevance in vivo are less extensive (only 2 of 8 figures). To infer behavioral significance, the authors generated a conditional GRPR knock-out mouse and restricted knock out to the NAc MSh using viral injection of Cre recombinase. This led to an increase in the break point in a progressive ratio lever pressing task. Although no attempts to study endogenous GRP release are reported, beyond reporting several GRP-expressing inputs to the NAc MSh, the increase in motivated lever pressing in KO mice advances our understanding GRPR's physiological significance in the NAc.

Major

1. The story lacks a model of how GRPR signaling might be coordinated to suppress motivated behavior. Such a model is limited by the lack of information about GRP-expressing inputs, yet some speculation is in order to guide future inquiries into this peptidergic signaling axis.
2. Did the authors make any attempts to study endogenous GRP release in brain slice, for example by optogenetically stimulating dopamine axons while recording from GRPR+ neurons? Even if unsuccessful, it would be worthwhile to report this in the SI. If successful, this would greatly enhance the study by providing insight into the physiologically relevant sources and how the activity patterns that drive release align with activity patterns observed in vivo (see point 1).
3. Did the authors attempt to use GRP-Light (the GRP sensor) during the operant conditioning task to detect endogenous release dynamics? The tool is readily available and could add depth to the in vivo component of the study.

Minor

4. The use of cFOS to measure GRP-evoked activity in vivo is not invalid but is not particularly insightful or compelling. It simply shows that it retains its excitatory properties in vivo. This could be relegated to the SI with more interesting in vivo data to show.
5. The introduction takes a long path to GRP in the NAc. Motivating the entire study by beginning with the broadly unknown functions of neuropeptides seems like an overly general place to start a study into this specific circuit. Are the authors suggesting that their approach is the best way to tackle the function of poorly understood neuropeptides, in general?
6. Do GRPR and D2R signaling interact in GRP+ MSNs, given that DA neurons are a primary source of GRP and that these two signaling pathways may oppose each other?

Reviewer #3

(Remarks to the Author)

Review of "Gastrin-releasing peptide signaling in the nucleus accumbens medial shell regulates neuronal excitability and motivation" from Aisenberg et al.

The authors present a very thorough investigation of Grpr+ neurons of the nucleus accumbens medial shell (NAc MSh). First they show that a subset of medium spiny neurons (MSNs) express Grpr in the striatum and determine that the majority of these neurons are part of the indirect pathway. Then they demonstrate these neurons are more likely to be spontaneously firing and have distinct excitability as compared to Grpr- MSNs. They also determine that the likely source of GRP is from glutamatergic neurons of the hippocampus, BLA or VTA. Finally, the authors deleted Grpr from NAc MSh and demonstrated a change in motivation as measured by the progressive ratio on an operant conditioning task. Altogether, I believe this is an

exhaustive characterization of neuronal population which plays a significant behaviour role. I believe that addressing the comments below could help improve this manuscript.

Comments:

Fig 1

1) In the text and figures, the authors sometime used Grpr::egfp or Grpr-egfp to refer to the transgenic line. Since this is a BAC transgenic and the "::" is usually for knockin, I would recommend sticking with Grpr-egfp.

Fig 2

2) The material and method state that 2-3 dots were needed to be Grpr+, yet there are example of cells in Fig a, c, d, and e were only 1 dot is visible.

Fig 4

3) The authors claim that absence of Grpr reduce the baseline excitability as Drd2+ MSN as measured by Fos mRNA. This claim should be supported by statistical analysis demonstrating a difference in % of Fos cells between WT and Grpr KO (e.g. Drd2+,Grpr+ WT VS KO). Also, it should be indicated which data points are males and females; this is particularly relevant since Grpr is on the X chromosome thus expression level could differ.

Fig 5

4) Why does way more Grp signal in "i" vs "j" ?

5) The sites of retrobeads injections should be illustrated.

6) This statement is not clear: "summed value from all slices in that region per 3 unique injection sites from 2 mice".

7) The authors claim that the VTA inputs are not glutamatergic, yet the transporter Slc17a6 should have used instead of Slc17a7.

Page 18

There is a small typo: Trinton-X-100 should be triton-X-100

Page 19

The full name of DPDMB should be provided.

Reviewer #4

(Remarks to the Author)

In this paper, Aisenberg et al. investigated the role of GRP signaling in the NAc in neuronal excitability and motivated behaviors.

GRP-GRPR pathway is known to be excitatory and has been reported to modulate behaviors in other brain regions. The authors found GRP-expressing cells in the NAc medial shell, an area well-known for receiving midbrain dopaminergic inputs. Whole-cell electrophysiology in Grpr-eGFP mice revealed that Grpr+ cells have high baseline excitability which is GRP signaling-dependent. FISH revealed 3 major clusters of Grpr-expressing cells, with the majority being iSPNs (both Drd2+/Penk+ and Drd2+/Penk-) and eSPNs (Casz1+). Further analyses with the Allen mouse and human brain scRNA-seq datasets largely agreed with the FISH result. The authors then compared the physiological properties between Grpr+ cells and the known SPN cell types using a clustering analysis, which revealed 3 groups of putative Grpr+ neurons, iSPNs, eSPNs, and immature SPNs.

The authors then connect GRP-GRPR signaling with neuronal activity in vivo. They began by comparing baseline Fos+ staining in Grpr+ (WT) and Grpr- (KO) mice and found that Grpr KO reduced the number of Fos+ cells, but only in Grpr+ iSPNs, rather than Grpr+ eSPNs. By performing Grp FISH in presynaptic regions of NAc, the authors found a few Grp-releasing possible presynaptic sites, and besides VTA, the other most abundant regions also express the excitatory neuron marker. Then the authors demonstrated the role of GRP on Grpr+ cells both in slice and in vivo. In slice, GRP infusion induced depolarization on Grpr+ cells, which could be blocked by GRPR antagonists. In vivo, unilateral infusion of GRP increased the number of Fos+ cells, specifically among the Grpr+ class. Lastly, at a behavior level, conditional deletion of Grpr in the NAc MSH improved motivation, as demonstrated by prolonged lever pressing in the Progressive Ratio task.

Overall, the experiments are well-performed, the analyses are appropriate, and the paper is well-written. I only have a few minor comments here:

1. DS Drd1+/Grpr+ neurons: The UMAP in Extended Data Fig. 5 b,d,f indicated the existence of DS Drd1+/Grpr+ clusters, which also agrees with Extended Data Fig. 4i. This might be worth mentioning in the discussion.
2. Figure 3: Physiological clustering: the authors stated, "Together, this analysis reveals multiple sub-populations of NAc GRPR+ neurons defined by their physiological properties, including putative iSPNs (cluster 1), eSPNs (cluster 0), and immature SPNs (cluster 3)." The logic of Figure 3 needs to be more clearly presented. As far as I understand, the current logic is, for a given sub-cluster of GRPR+ neurons, if the physiological property is not significantly different from the majority of another canonical population, then this GRPR+ sub-cluster belongs to that canonical population. Using that logic, since D1 and D2 cells don't differ in all these measurements, should you say D1 and D2 cells belong to the same category? Also, since there was no direct electrophysiological data from eSPNs, it is hard to claim that cluster 0 are eSPNs (I only get that cluster 0 are not iSPNs and not immature cells).

3. Figure 6i: the best control experiment should be Grp injection in Grpr knock-out animals, which is worth discussing at least. Also, there was no histology for Grp infusion experiment.
4. Figure 7: the current PR ratio phenotype aligns well with what we know about D2-SPNs, but what about eSPNs and immature SPNs, which also seemed to be targeted according to Fig 7b? Along the same line, it would be helpful to perform Grpr ISH to quantify the degree of Grpr Exon 2 removal in D2-SPN and eSPN. Also, it would be good to introduce what is known about the eSPN's physiological and functional properties when it is first mentioned in the text.
5. Page 7, "we used multi-plex FISH with markers for known striatal cell types to determine their identify (Fig. 2a-j)" here "identify" should be "identity".

Version 1:

Reviewer comments:

Reviewer #1

(Remarks to the Author)

In this revision, the authors thoughtfully conducted new experiments in response to all of my comments and sincerely addressed each of them based on the results. I feel grateful for this, and I hope that such excellent research will be published in this journal soon, making a significant contribution to the NAc field.

(Remarks on code availability)

For some reason, the access to the link was prohibited.

Reviewer #2

(Remarks to the Author)

The authors have adequately address my primary concerns.

(Remarks on code availability)

Reviewer #3

(Remarks to the Author)

All my comments have been addressed.

(Remarks on code availability)

Reviewer #4

(Remarks to the Author)

The authors have satisfied my concerns

(Remarks on code availability)

Response to reviewers for Aisenberg et al, NCOMMS-24-64113-T

We thank the editor and reviewers for taking the time to evaluate our manuscript and provide constructive feedback. We have added several new experiments, analyses and text revisions to address these comments as outlined below. We believe the study is strongly improved and hope you will now find it suitable for publication in Nature Communications.

Responses to specific reviewer comments:

Reviewer #1 (Remarks to the Author):

Aisenberg and colleagues identified previously undescribed subtypes of spiny projection neurons (SPNs) in the medial shell of the nucleus accumbens (NAc MSh) by using single-cell RNA sequencing and fluorescent in situ hybridization (FISH). They also characterized the electrophysiological properties of Grpr+ neurons. Increased Fos+ labeling supports the electrophysiological findings. Their role in controlling motivated behavior was demonstrated. These findings are a valuable contribution to the understanding of the complex architecture of the ventral striatum. The amount of work is substantial, and the findings provide detailed information on diverse cell types within the NAc and establish molecular markers for Grpr neurons. However, to strengthen the findings, more rigorous electrophysiological recordings and additional approaches to confirm in vivo function will be needed as described below.

We appreciate the reviewer's comments that our findings "are a valuable contribution to the understanding of the complex architecture of the ventral striatum" and that "the amount of work is substantial, and the findings provide detailed information on diverse cell types within the NAc"

Is it possible to examine the Grpr::egfp with D2 mice line to confirm that these are indeed D2-expressing neurons?

The D2 mouse line that is available is a *Drd2*-GFP line; therefore, it is not possible to use this together with the *Grpr*-eGFP mice. We instead bred the *Grpr*-eGFP mice to D1-tdTomato mice and showed lack of co-localization of *Grpr*-eGFP with D1-tdTomato in the NAc in Extended Data Fig. 2i-k. Since ~95% of striatal neurons are D1 or D2-expressing neurons, we infer that if the cells are not D1-tdTomato+, then they are likely to be D2+ neurons or possibly interneurons. Since interneurons in the striatum have few to no spines, and all *Grpr*-eGFP+ cells have spines (see Extended Data Fig. 2f-h), we conclude that they are D2 SPNs.

To support this claim, we have bred the *Grpr*-eGFP mice to an *Adora2a*-Cre mouse line together with the Ai9 Cre-dependent tdTomato reporter line. *Adora2a*-Cre mice label adenosine 2a receptor-expressing cells and have been validated to label D2-expressing iSPNs (Gong, S. et al, *J Neurosci*, 2007). We find ~93% colocalization of *Grpr*-eGFP and tdTomato in the NAc MSh of these mice. We have added this data to the new Extended Data Fig. 2 (panels I-n, see also Fig. R1).

Fig. R1: The majority of *Grpr*-eGFP+ neurons in the NAc MSh express *Adora2a* (A2AR).
(a) Confocal image of the NAc

Msh from a *Grpr-eGFP;Adora2a-Cre;Ai9* mouse. Cre-expressing cells are labeled with tdTomato (magenta). **(b)** Zoomed-in images of the boxed region showing GFP+ cells that are also tdTomato+. **(c)** Mean \pm SEM percent of GFP+ cells that are tdTomato+ in the NAc MSh. n=2 mice (2-3 sections were quantified per mouse).

Does membrane potential In Figure 1 and Expanded data 2 mean resting membrane potential? How was resting membrane potential measured? Description in the method section is too simple.

Yes, this refers to resting membrane potential, we have clarified this in the results and figure legends. We have added the following sentence to the methods: “Resting membrane potential (Vm) was calculated from current clamp traces recorded for the input-output experiment. Vm was calculated as the average membrane potential over a 300 ms baseline period between 50-350 ms of a two second recording. The plotted value for resting membrane potential is the average of this value for all traces over a two-minute period”.

In page 7, “Given that *Grpr-eGFP*+ cells were spontaneously active..”, If hyperpolarizing current is injected to lower the membrane potential of *Grpr-eGFP*+ neurons, do these neurons continue to fire spontaneously?

From our analysis, it is likely that the *Grpr-eGFP* cells are spontaneously active due to their depolarized membrane potential. If we analyze the probability of a cell firing spontaneously upon break-in versus their resting membrane potential, we find that no cells with membrane potentials below -60 mV were spontaneously active (**Fig. R2**). Therefore, if we were to hyperpolarize the membrane beyond that level, we would not expect the cells to fire spontaneously. Spontaneous synaptic activity is generally very low in striatal slices; therefore, we do not expect that the spontaneous firing is driven by synaptic inputs. Consistent with this, when spontaneous activity was observed in *Grpr-eGFP*+ neurons it was rhythmic in nature, which is inconsistent with stochastic synaptic activation.

Fig. R2: Spontaneous activity of *Grpr-eGFP*+ neurons relates to resting membrane potential. (a) Graph of the percentage of spontaneously active *Grpr-eGFP* cells according to their resting membrane potential. **(b)** The same data as in **a**, plotted as the number of cells that were spontaneously active in each 5 mV bin of resting membrane potential. “n” refers to the number of cells in each bin.

The sample traces in Extended Data Fig. 3a are identical to those in Fig. 1. Please provide a different example.

We initially included the same sample traces, as the *Grpr*-eGFP data in the current Extended Data Fig. 5 is the same as in Figure 1 (as stated in the figure legend). We have now replaced the sample traces in Extended Data Fig. 5a to show different examples.

To better assess recording quality, please show how the traces look like at each step of current injection throughout the 0-100 mV range (Fig. 11).

We have added example traces for each current step for each recorded cell type in the new Extended Data Fig. 3. See also **Fig. R3** below.

Fig. R3: Example traces in response to positive current steps for different NAc MSh cell types. (a-e) Example traces elicited by the indicated current steps for a representative *Grpr*-eGFP+ neuron (green, a), *Grpr*-eGFP- neuron (black, b), D1-tdTomato+ neuron (magenta, c), D1-tdTomato-;*Grpr*-eGFP- neuron (blue, d), and ChAT-Cre+ neuron (cyan, e). *This is the new Extended Data Fig. 3 in the manuscript.

In Figure 4, the representative image shows only a few Fos+ neurons. Counting such a small number in each sample window may lead to biased results. To obtain a more accurate representation of the population, please show and analyze bigger representative images from each sample (i.e., 250 x 250 μm), providing a clearer view of whether the majority of Grpr+ neurons are Fos+ neurons at the population level.

We apologize that this wasn't clear in the original manuscript. For all cell counting experiments (Figures 2, 3, 4, 5, 6, E1, E2, E6, E10, E13, and E14), we quantified all cells within the NAc medial shell. For example, for Figure 4, we counted hundreds to thousands of neurons in the NAc MSh per mouse. We have clarified this in the methods section and figure legends. We have also added the total number of neurons counted to the figure legend for Figure 4.

What percentage of Fos+ neurons are Grpr+ neurons?

We have analyzed this and included it in the revised Figure 4 (see also Fig. R4). The *Grpr*+ cells represent ~4% of the Fos+ population in the NAc MSh, consistent with their relative sparsity.

Fig. R4: Percentage of Fos+ cells in the NAc MSh that are Grpr+. Mean ± SEM percentage of Fos+ (n=8 mice) in the NAc MSh that are Grpr+. Dots represent values for individual mice.

In Figure 4, please show the number of neurons analyzed per mouse as well as the total number of neurons.

The number of neurons analyzed per mouse has been added to the figure legend. See also the legend for Fig. R8 below.

Was there a decrease in Fos+ neurons in Grpr KO mice?

We have quantified this and do not find a significant difference in the total number of Fos+ cells in the NAc MSh in *Grpr* KO mice (Fig. R5). This analysis is included in the revised Extended Data Fig. 10 and also shown here. This is consistent with the fact that *Grpr*+ cells make up only a small fraction of the total Fos+ cells in this region (see Fig. R4).

Fig. R5: There is no difference in the total number of Fos+ cells in the NAc MSh between Grpr WT and KO mice. Mean +/- SEM total number of Fos+ cells in the NAc MSh in WT (black) and *Grpr* KO (red) mice. (n=4 WT and n=7 KO mice, p=0.4121, Mann-Whitney test). Dots represent values for individual mice.

In Figure 6, test whether GRP application still increases membrane potential at -80 mV when hyperpolarizing current is injected. If it does, by how much does the membrane potential

increase?

We have re-analyzed our original dataset and plotted the change in membrane potential following GRP wash-on as a function of the cell's resting membrane potential (**Fig. R6**). We find that 9 out of 10 cells depolarized in response to GRP, and there is no significant relationship between the magnitude of the response and the resting membrane potential ($r^2=0.2947$, $p=0.1049$, simple linear regression). While we do not have any cells that had a resting membrane potential below -70mV, we find that cells with a membrane potential between -60 and -70mV depolarized, on average, by 8.3 mV.

Fig. R6: Change in membrane potential after GRP wash-on versus resting membrane potential. Plot showing the change in membrane potential versus the resting membrane potential for 10 *Grpr*-eGFP+ neurons (dots). $r^2=0.2947$, $p=0.1049$, simple linear regression.

Please include control experiments; GRP application in *Grpr*-lineage cells in *Grpr* KO mice.

We have not been able to breed the *Grpr*-eGFP mice together with the *Grpr* KO mice, which would be required for a slice electrophysiology experiment with GRP application. To address this comment, we have instead injected GRP into the NAc MSh of WT and *Grpr* KO mice *in vivo* (**Fig. R7**). In this dataset, we replicated our original finding that in WT mice, GRP injection significantly increases the percentage of *Grpr*+ cells that are *Fos*+ without affecting the total number of *Grpr*+ cells. We find no significant change in the percentage of *Grpr*-lineage cells that are *Fos*+ in *Grpr* KO mice, confirming that this effect requires signaling through GRPR. This data is included in the new Extended Data Figure 13.

requires signaling through GRPR. This data is included in the new Extended Data Figure 13.

Fig. R7: *Grpr* KO mice do not have increased *Fos* expression in *Grpr*-lineage cells in response to GRP injection *in vivo*. (a-b) Representative images of the NAc MSh from *Grpr* WT mice from the (a) injected hemisphere (3 μ M GRP) and (b) un-injected hemisphere. FISH for *Grpr* in magenta and *Fos* in cyan. Nuclei are labeled with DAPI in blue. Dashed circles outline *Grpr*⁺ cells, white=*Grpr*⁺/*Fos*⁻, cyan=*Grpr*⁺/*Fos*⁺. (c) Percentage of *Grpr*⁺ cells in the NAc MSh that were *Fos*⁺ in the un-injected vs GRP-injected hemisphere of *Grpr* WT mice. (n=5 mice, ***p=0.0007, paired t-test). (d) Total number of *Grpr*⁺ cells in the NAc MSh in the un-injected vs GRP-injected hemisphere of *Grpr* WT mice (n=5 mice, p=0.7500, Wilcoxon test). (e-f) Same as (a-b) but in littermate *Grpr* KO mice. (g) Percentage of *Grpr*-lineage cells in the NAc MSh that were *Fos*⁺ in the un-injected vs GRP-injected hemisphere of *Grpr* KO mice (n=8 mice, p=0.4000, paired t-test). (h) Total number of *Grpr*-lineage cells in the NAc MSh in the un-injected vs GRP-injected hemisphere of *Grpr* KO mice (n=8 mice, p=0.3828, Wilcoxon test). For panels c-d and g-h, data were summed across two sections per mouse. Dots represent values for individual mice. *This is the new Extended Data Fig. 13 in the manuscript.

Inject Grp into *Grpr* KO mice to determine whether *Fos*⁺ neurons are not increasing

Please see our response to the comment above, we have performed this experiment and find no change in the percentage of *Grpr*-lineage cells that are *Fos*⁺ in *Grpr* KO mice (**Fig. R7**).

Discuss potential scenarios in which the membrane potential remains unchanged while the membrane resistance decreases.

We believe the reviewer is referring to the data in Extended Data Fig. 2 (now Extended Data Fig. 4) in which we found a significant reduction in membrane resistance with DPDMB application to *Grpr*⁺ cells but only a downward trend in the membrane potential. As noted by the reviewer, input resistance and resting membrane potential are often related since they are both influenced by potassium channels that are open at rest. However, we found that applying the GRPR antagonist DPDMB significantly reduced the average membrane resistance by ~220 MOhms, while the average resting membrane potential did not change significantly (although this trended downward). One possibility is that outward rectifying K channels are responsible for setting the resting membrane potential, which means that the cell will not be able to hyperpolarize more when these channels are active but that the cell would depolarize with application of the agonist. We assume that given the high input resistance of these cells, small decreases in leak conductance have a relatively smaller impact on *V_m*. Note that the larger changes in input resistance associated with GRP wash-on led to correlated increases in membrane resistance and a depolarized membrane potential.

In page 8, figure number is missing, (Extended Data Fig. c-f).

Thank you for pointing this out, this has been corrected.

Reviewer #2 (Remarks to the Author):

This manuscript primarily examines the characteristics of medium spiny neurons in the nucleus accumbens medial shell that express the gastrin-releasing peptide receptor. Using a combination of RNASeq and FISH, the authors found that there are distinct populations of Penk-lacking D2-MSNs in the NAc mSh that express GRPR. They extensively characterize these neurons in terms of their molecular identity and electrophysiological properties. A major conclusion is that GRPR+ D2-MSNs to comprise a subset of eccentric MSNs. The cellular characterization is excellent (6 of 8 figures). The authors efforts to establish physiological relevance in vivo are less extensive (only 2 of 8 figures). To infer behavioral significance, the authors generated a conditional GRPR knock-out mouse and restricted knock out to the NAc MSh using viral injection of Cre recombinase. This led to an increase in the break point in a progressive ratio lever pressing task. Although no attempts to study endogenous GRP release are reported, beyond reporting several GRP-expressing inputs to the NAc MSh, the increase in motivated lever pressing in KO mice advances our understanding GRPR's physiological significance in the NAc.

We thank the reviewer for their feedback and appreciate the positive comments, in particular that the “cellular characterization is excellent”.

Major

1. The story lacks a model of how GRPR signaling might be coordinated to suppress motivated behavior. Such a model is limited by the lack of information about GRP-expressing inputs, yet some speculation is in order to guide future inquiries into this peptidergic signaling axis.

We have expanded upon this point in the discussion and speculated on how GRP-release from the VTA, ventral hippocampus, subiculum, and BLA inputs to the NAc MSh could serve to counteract or constrain the positive effects of these inputs on reward-related motivated behavior.

2. Did the authors make any attempts to study endogenous GRP release in brain slice, for example by optogenetically stimulating dopamine axons while recording from GRPR+ neurons? Even if unsuccessful, it would be worthwhile to report this in the SI. If successful, this would greatly enhance the study by providing insight into the physiologically relevant sources and how the activity patterns that drive release align with activity patterns observed in vivo (see point 1).

We agree that this is an interesting experiment although we note that it is typically difficult to evoke neuropeptide release using optogenetics. We had obtained a GRP-Cre mouse line to attempt to evoke GRP release from all inputs to the NAc but we found that the mouse line under-labeled GRP expressing cells. For example, the mouse had no Cre expression in the VTA, which we have shown has robust expression of *Grp* mRNA. In addition, we could not observe any labeled inputs in the NAc MSh when the GRP-Cre mouse was crossed to a tdTomato Cre reporter. Given limitations of the mouse tools and the fact that no one has previously demonstrated endogenous GRP release in the brain, we feel this experiment is beyond the scope of the current study.

3. Did the authors attempt to use GRP-Light (the GRP sensor) during the operant conditioning

task to detect endogenous release dynamics? The tool is readily available and could add depth to the *in vivo* component of the study.

While this is an interesting experiment, our lab is not set-up to perform or analyze *in vivo* recordings during behavior in freely moving animals. Through personal communication with Dr. Bernardo Sabatini, who has used the grpLight sensor (Melzer, S. et al, Cell, 2021), we believe that the sensitivity of the published sensor may not be sufficient to detect endogenous release in a behaviorally relevant context. Since there is no published precedence for this experiment, we feel it is beyond the scope of the study.

Minor

4. The use of cFOS to measure GRP-evoked activity *in vivo* is not invalid but is not particularly insightful or compelling. It simply shows that it retains its excitatory properties *in vivo*. This could be relegated to the SI with more interesting *in vivo* data to show.

We believe that this is important data that shows that GRP activates a select ensemble of NAc neurons *in vivo*. We prefer to keep it as part of the main figure but could move it to Extended Data if needed.

5. The introduction takes a long path to GRP in the NAc. Motivating the entire study by beginning with the broadly unknown functions of neuropeptides seems like an overly general place to start a study into this specific circuit. Are the authors suggesting that their approach is the best way to tackle the function of poorly understood neuropeptides, in general?

We appreciate the reviewer's point and have revised the introduction accordingly. Our goal is not to study neuropeptides in general, but rather comprehensively characterize the genetic and cellular properties of *Grpr*-expressing neurons in the NAc MSh, which represent a previously unreported cell type.

6. Do GRPR and D2R signaling interact in GRP+ MSNs, given that DA neurons are a primary source of GRP and that these two signaling pathways may oppose each other?

This is an interesting possibility. We have added comments related to the possible interactions of GRP and dopamine within the NAc MSh to the discussion.

Reviewer #3 (Remarks to the Author):

Review of "Gastrin-releasing peptide signaling in the nucleus accumbens medial shell regulates neuronal excitability and motivation" from Aisenberg et al.

The authors present a very thorough investigation of *Grpr*+ neurons of the nucleus accumbens medial shell (NAc MSh). First they show that a subset of medium spiny neurons (MSNs) express *Grpr* in the striatum and determine that the majority of these neurons are part of the indirect pathway. Then they demonstrate these neurons are more likely to be spontaneously firing and have distinct excitability as compared to *Grpr*- MSNs. They also determine that the likely source of GRP is from glutamatergic neurons of the hippocampus, BLA or VTA. Finally, the

authors deleted *Grpr* from NAc MSh and demonstrated a change in motivation as measured by the progressive ratio on an operant conditioning task. Altogether, I believe this is an exhaustive characterization of neuronal population which plays a significant behaviour role. I believe that addressing the comments below could help improve this manuscript.

We thank the reviewer for their evaluation and positive comments.

Comments:

Fig 1

1) In the text and figures, the authors sometime used *Grpr::egfp* or *Grpr-egfp* to refer to the transgenic line. Since this is a BAC transgenic and the “::” is usually for knockin, I would recommend sticking with *Grpr-egfp*.

We agree with the reviewer and have revised the nomenclature to *Grpr-eGFP* throughout the manuscript.

Fig 2

2) The material and method state that 2-3 dots were needed to be *Grpr+*, yet there are example of cells in Fig a, c, d, and e where only 1 dot is visible.

We have revised the methods to clarify that cells were considered *Grpr+* if the in situ puncta signal covered >1% of their nuclear area, defined by DAPI.

Fig 4

3) The authors claim that absence of *Grpr* reduce the baseline excitability as *Drd2+* MSN as measured by *Fos* mRNA. This claim should be supported by statistical analysis demonstrating a difference in % of *Fos* cells between WT and *Grpr* KO (e.g. *Drd2+*, *Grpr+* WT VS KO).

We performed the analysis suggested by the reviewer and included this in the revised Figure 4 (**Fig. R8**). We find that in *Grpr* KO mice, there is no longer a difference in the percentage of *Fos+* cells between *Drd2+* cells that do or don't co-express *Grpr*. We note that we did not see a significant difference in the *Fos+* positivity of *Grpr+/Drd2+* cells between WT and KO mice. However, we did find a significant difference in the percentage of *Fos+* cells in the *Grpr+/Penk+* sub-population between WT and KO mice. Notably, there was no difference between WT and KO mice for the *Grpr+/Casz1+* population. Since the *Grpr+/Drd2+* cell population contains both *Penk+* and *Casz1+* cells, this can explain why there was no significant difference in the overall *Grpr+/Drd2+* population with *Grpr* deletion.

Fig. R8: Deletion of *Grpr* decreases the number of *Fos+*/*Grpr+*/*Penk+* cells. (a) Mean \pm SEM percentage of *Drd2+*/*Grpr+* (magenta) vs. *Drd2+*/*Grpr-* (green) cells in the NAc MSh of WT mice (left) and *Grpr* KO mice (right) that are *Fos+*. (WT: n=8 mice, 517-1215 neurons counted per mouse, 6849 total neurons counted; KO: n=7 mice, 246-2125 neurons counted per mouse, 9340 total neurons counted); **p=0.0018, one-way ANOVA; Kruskal-Wallis with Dunn's correction for multiple comparisons, **p=0.0013 *Drd2+*/*Grpr+* WT vs *Drd2+*/*Grpr-* WT, p=0.3646 *Drd2+*/*Grpr+* *Grpr* KO vs *Drd2+*/*Grpr-* *Grpr* KO, p=0.6958 *Drd2+*/*Grpr+* WT vs *Drd2+*/*Grpr+* *Grpr* KO). Dots represent values for individual mice. **(b)** Mean \pm SEM percentage of *Penk+*/*Grpr+* (magenta) vs. *Penk+*/*Grpr-* (green) cells in the NAc MSh of WT mice (left) and *Grpr* KO mice (right) that are *Fos+*. (WT: n=6 mice, 534-955 neurons counted per mouse, 3941 total neurons counted; KO: n=8 mice, 396-1359 neurons counted per mouse, 6916 total neurons counted); ***p=0.0008, one-way ANOVA; Holm-Sidak's multiple comparisons, **p=0.0025 *Penk+*/*Grpr+* WT vs *Penk+*/*Grpr-* WT, p=0.8555 *Penk+*/*Grpr+* *Grpr* KO vs *Penk+*/*Grpr-* *Grpr* KO, **p=0.0029 *Penk+*/*Grpr+* WT vs *Penk+*/*Grpr+* *Grpr* KO). **(c)** Mean \pm SEM percentage of *Casz1+*/*Grpr+* (magenta) vs. *Casz1+*/*Grpr-* (green) cells in the NAc MSh of WT mice (left) and *Grpr* KO mice (right) that are *Fos+*. (WT: n=8 mice, 442-970 neurons counted per mouse, 5117 total neurons counted; KO: n=5 mice, 223-606 neurons counted per mouse, 1690 total neurons counted); ***p=0.0002, one-way ANOVA; Holm-Sidak's multiple comparisons, **p=0.0021 *Casz1+*/*Grpr+* WT vs *Casz1+*/*Grpr-* WT, **p=0.0024 *Casz1+*/*Grpr+* *Grpr* KO vs *Casz1+*/*Grpr-* *Grpr* KO, p=0.6717 *Casz1+*/*Grpr+* WT vs *Casz1+*/*Grpr+* *Grpr* KO. Values were obtained by counting all cells in the NAc MSh from two independent slices per mouse that were summed together.

Also, it should be indicated which data points are males and females; this is particularly relevant since *Grpr* is on the X chromosome thus expression level could differ.

We thank the reviewer for pointing this out. We have now included analysis of the percentage of *Grpr+* cells by sex and the expression level of *Grpr* mRNA per cell by sex. We find no significant sex differences. This data is included in the new Extended Data Figure 1 and also below (**Fig. R9**). We have indicated in the figure legends if the experiment was performed in males and females (the majority of experiments) or in males only. The only experiments that used males only were the *Grpr* knock-out mouse experiments in Figures 4, Extended Data Fig. 10 and Extended Data Fig. 13.

Fig. R9: NAc MSh *Grpr* expression does not differ by sex. (a) Mean \pm SEM total number of *Grpr+* cells in the NAc MSh by sex (male=blue, female=pink). (n=11 male and 9 female mice; p=0.8972, Mann-Whitney test). Dots represent values for individual mice. **(b)** Scatterplots show the percent nuclear area covered by *Grpr* mRNA puncta in *Grpr*-expressing NAc MSh neurons for male and female mice. Lines represent the median. n=1339 cells from 11 male mice and n=1089 cells from 9 female mice; p=0.1097, unpaired t-test.

Fig 5

4) Why does way more Grp signal in "i" vs "j" ?

These experiments were performed in two batches and we found that the sections processed in the first batch (*Grp* and *Slc17a7*) had overall brighter signal compared to the second (*Grp* and *Gad2*). We don't believe this impacts the results as we detected a similar number of *Grp+* cells

in each batch and quantification was performed within each batch and not compared across batches.

5) The sites of retrobeads injections should be illustrated.

We appreciate this suggestion. Images of the retrobead injection sites are included in the new Extended Data Figure 11 (see also **Fig. R10**).

Fig. R10: Images of retrobead injection sites in the NAc MSh. (a-d) Images showing red or green fluorescent retrobead injection sites in the NAc MSh with DAPI staining in blue. Four independent injections were made across three mice.

6) This statement is not clear: “summed value from all slices in that region per 3 unique injection sites from 2 mice”.

We have revised this to state that “dots represent values for three independent injection sites”. For this experiment, we had one mouse that was injected with red retrobeads in one hemisphere and green retrobeads into the other hemisphere (see **Fig. R10c**). We found no cross hemisphere transport of the retrobeads, therefore these were considered independent injections. In the other mice, we performed unilateral injections with red retrobeads only.

7) The authors claim that the VTA inputs are not glutamatergic, yet the transporter *Slc17a6* should have used instead of *Slc17a7*.

We thank the reviewer for pointing this out. We agree that *Slc17a7* is not the optimal marker to determine whether the VTA *Grp+* cells are glutamatergic. We have removed this data from Figure 5 and removed any statements about whether the VTA inputs are glutamatergic.

Page 18

There is a small typo: Trinton-X-100 should be triton-X-100

This has been fixed.

Page 19

The full name of DPDMB should be provided.

We have included the full name of DPDMB: (D-Phe⁶,Leu-NHET¹³,des-Met¹⁴)-Bombesin (6-14)

Reviewer #4 (Remarks to the Author):

In this paper, Aisenberg et al. investigated the role of GRP signaling in the NAc in neuronal excitability and motivated behaviors.

GRP-GRPR pathway is known to be excitatory and has been reported to modulate behaviors in other brain regions. The authors found GRP-expressing cells in the NAc medial shell, an area well-known for receiving midbrain dopaminergic inputs. Whole-cell electrophysiology in Grpr-eGFP mice revealed that Grpr+ cells have high baseline excitability which is GRP signaling-dependent. FISH revealed 3 major clusters of Grpr-expressing cells, with the majority being iSPNs (both Drd2+/Penk+ and Drd2+/Penk-) and eSPNs (Casz1+). Further analyses with the Allen mouse and human brain scRNA-seq datasets largely agreed with the FISH result. The authors then compared the physiological properties between Grpr+ cells and the known SPN cell types using a clustering analysis, which revealed 3 groups of putative Grpr+ neurons, iSPNs, eSPNs, and immature SPNs.

The authors then connect GRP-GRPR signaling with neuronal activity in vivo. They began by comparing baseline Fos+ staining in Grpr+ (WT) and Grpr- (KO) mice and found that Grpr KO reduced the number of Fos+ cells, but only in Grpr+ iSPNs, rather than Grpr+ eSPNs. By performing Grp FISH in presynaptic regions of NAc, the authors found a few Grp-releasing possible presynaptic sites, and besides VTA, the other most abundant regions also express the excitatory neuron marker. Then the authors demonstrated the role of GRP on Grpr+ cells both in slice and in vivo. In slice, GRP infusion induced depolarization on Grpr+ cells, which could be blocked by GRPR antagonists. In vivo, unilateral infusion of GRP increased the number of Fos+ cells, specifically among the Grpr+ class. Lastly, at a behavior level, conditional deletion of Grpr in the NAc MSh improved motivation, as demonstrated by prolonged lever pressing in the Progressive Ratio task.

Overall, the experiments are well-performed, the analyses are appropriate, and the paper is well-written.

We thank the reviewer for their comments and appreciate their positive assessment.

I only have a few minor comments here:

1. DS Drd1+/Grpr+ neurons: The UMAP in Extended Data Fig. 5 b,d,f indicated the existence of DS Drd1+/Grpr+ clusters, which also agrees with Extended Data Fig. 4i. This might be worth mentioning in the discussion.

We agree that this is an interesting point. Indeed, we do observe a population of Grpr+ neurons in the dorsal striatum that co-express Drd1. Interestingly, we generally do not find many Grpr+/Drd1+ cells in the ventral striatum, suggesting that GRP may modulate the activity of different populations of cells in the DS and VS. We have added a few sentences to the discussion regarding this point.

2. Figure 3: Physiological clustering: the authors stated, “Together, this analysis reveals multiple sub-populations of NAc GRPR+ neurons defined by their physiological properties, including putative iSPNs (cluster 1), eSPNs (cluster 0), and immature SPNs (cluster 3).” The logic of Figure 3 needs to be more clearly presented. As far as I understand, the current logic is, for a given sub-cluster of GRPR+ neurons, if the physiological property is not significantly different from the majority of another canonical population, then this GRPR+ sub-cluster belongs to that canonical population. Using that logic, since D1 and D2 cells don’t differ in all these measurements, should you say D1 and D2 cells belong to the same category? Also, since there was no direct electrophysiological data from eSPNs, it is hard to claim that cluster 0 are eSPNs (I only get that cluster 0 are not iSPNs and not immature cells).

We thank the reviewer for this comment. We have revised this section in the results to provide more clarity. When analyzing D1 and D2 SPNs together with the three sub-types of GRPR+ cells, we do find that the properties of D1 and D2 cells are similar to each other. The relatively hyperpolarized membrane potential, lower membrane resistance, and shorter AP half-width represent a “canonical SPN” signature (although we and others have reported subtle differences between the D1 and D2 SPN populations in terms of their intrinsic properties). Within the GRPR sub-populations, defined by unbiased clustering, we find that the cluster 1 cells are most similar to canonical SPNs, and don’t differ significantly across the parameters measured. We therefore conclude that these may be GRPR+ iSPNs. By contrast, clusters 0 and 3 were significantly different from canonical SPNs and clustered out in the UMAP. We recorded from immature iSPNs and found that the adult GRPR+ cluster 3 cells were similar in their properties to immature cells and concluded that these might represent GRPR+ iSPNs that are in an immature state. The GRPR cluster 0 cells were distinct from both mature and immature iSPNs, therefore based on the genetic sub-types we defined, we suggest that these could be GRPR+ eSPNs. The reviewer is correct that we cannot conclusively claim that these are eSPNs as there is no published analysis of eSPN properties. We have stated this in the results. This would likely require a reporter mouse that expresses eSPN markers such as *Casz1* or *Otof*. To our knowledge, these mice have not yet been established.

3. Figure 6i: the best control experiment should be Grp injection in Grpr knock-out animals, which is worth discussing at least. Also, there was no histology for Grp infusion experiment.

We have performed the experiment suggested by the reviewer, to inject GRP into *Grpr* KO mice. In this experiment, we replicated our original finding that GRP increases *Fos* expression in *Grpr*+ cells in WT mice. We find that this does not occur in *Grpr* KO mice. We conclude that GRPR is required for the GRP-induced activation of *Grpr*-lineage cells in the NAc MSh. These data are included in the new Extended Data Fig. 13 and provided above in **Fig. R7**.

During this GRP injection experiment, we injected an additional WT mouse with AAV-GFP using the same coordinates to show the location of the infusion. We are including this image here for the reviewer’s reference (**Fig. R11**). Since we don’t know how far the GRP spreads *in vivo* compared to AAV-GFP, we did not include this in the main manuscript.

Fig. R11: Approximate injection site for the GRP infusion experiments. Confocal image of the dorsal striatum (DS) and NAc MSh from a wild-type mouse injected with AAV-GFP into the MSh. GFP expression is in grey. The

same injection coordinates (M/L +0.65 mm, A/P +1.90 mm, D/V -4.50 mm) and injection volume (800 nl) were used for the GRP injections.

4. Figure 7: the current PR ratio phenotype aligns well with what we know about D2-SPNs, but what about eSPNs and immature SPNs, which also seemed to be targeted according to Fig 7b?

In prior studies that targeted D2 SPNs, *Drd2*-Cre mice were used, therefore the manipulations would have likely targeted canonical D2 iSPNs, as well as *Drd2* expressing immature SPNs and putative eSPNs. Therefore, the individual contribution of these other cell types to the previously reported behaviors is currently unknown. We agree that in our *Grpr* KO mice, all three types of D2 SPNs would also be targeted. Therefore, future studies using intersectional genetics approaches (i.e. with Cre and Flp) are likely needed to selectively isolate the contributions of the different D2 SPN populations in the NAc MSh.

Along the same line, it would be helpful to perform *Grpr* ISH to quantify the degree of *Grpr* Exon 2 removal in D2-SPN and eSPN.

We agree that this would be helpful; however, the only probe that is available from RNAScope to measure *Grpr* mRNA does not detect the removal of exon 2. This would require the development of a custom probe. We expect that the recombination efficiency of the floxed *Grpr* allele following exposure to Cre should not differ among different cell types. Therefore, all *Grpr*-expressing cells should be targeted in the conditional KO experiments.

Also, it would be good to introduce what is known about the eSPN's physiological and functional properties when it is first mentioned in the text.

We agree, however, nothing is currently known about the physiological and functional properties of eSPNs since there is not yet a mouse reporter line that selectively labels them. Thus far, they have only been described in terms of their gene expression profile in scRNA-sequencing experiments. In the introduction we state that "Unlike dSPNs and iSPNs, nothing is known about the functional properties of eSPNs as there is currently no mouse line that selectively labels them."

5. Page 7, "we used multi-plex FISH with markers for known striatal cell types to determine their identify (Fig. 2a-j)" here "identify" should be "identity".

The typo has been fixed.

Response to reviewers for NCOMMS-24-64113A

We thank the editor and reviewers for considering our manuscript. We were pleased to see that there were no further concerns raised by the reviewers.

Reviewer #1 (Remarks to the Author):

In this revision, the authors thoughtfully conducted new experiments in response to all of my comments and sincerely addressed each of them based on the results. I feel grateful for this, and I hope that such excellent research will be published in this journal soon, making a significant contribution to the NAc field.

Reviewer #1 (Remarks on code availability):

For some reason, the access to the link was prohibited.

The GitHub repository has been made public and the code should be accessible.

Reviewer #2 (Remarks to the Author):

The authors have adequately address my primary concerns.

We thank the reviewer for their time.

Reviewer #3 (Remarks to the Author):

All my comments have been addressed.

We thank the reviewer for their time.

Reviewer #4 (Remarks to the Author):

The authors have satisfied my concerns

We thank the reviewer for their time.